# MULTI-AGENT DEEP REINFORCEMENT LEARNING UNDER CONSTRAINED COMMUNICATIONS

## ABSTRACT

Centralized training with decentralized execution (CTDE) has been the dominant paradigm in multi-agent reinforcement learning (MARL), but its reliance on global state information during training introduces scalability, robustness, and generalization bottlenecks. Moreover, in practical scenarios such as adding/dropping teammates or facing environment dynamics that differ from the training, CTDE methods can be brittle and costly to retrain, whereas distributed approaches allow agents to adapt using only local information and peer-to-peer communication. We present a distributed MARL framework that removes the need for centralized critics or global information. Firstly, we develop a novel Distributed Graph Attention Network (D-GAT) that performs global state inference through multi-hop communication, where agents integrate neighbor features via input-dependent attention weights in a fully distributed manner. Leveraging D-GAT, we develop the distributed graph-attention MAPPO (DG-MAPPO) – a distributed MARL framework where agents optimize local policies and value functions using local observations, multi-hop communication, and shared/averaged rewards. Empirical evaluation on the StarCraftII Multi-Agent Challenge, Google Research Football, and Multi-Agent Mujoco demonstrates that our method consistently outperforms strong CTDE baselines, achieving superior coordination across a wide range of cooperative tasks with both homogeneous and heterogeneous teams. Our distributed MARL framework provides a principled and scalable solution for robust collaboration, eliminating the need for centralized training or global observability. To the best of our knowledge, DG-MAPPO appears to be the first to fully eliminate reliance on privileged centralized information, enabling agents to learn and act solely through peer-to-peer communication.

## 1 INTRODUCTION

Multi-agent reinforcement learning (MARL) has emerged as a powerful framework for training multiple agents to learn cooperative and competitive behaviors in complex dynamic environments (Zhang et al., 2021). However, learning effective collaborative policies remains challenging, as each agent simultaneously seeks to maximize its own return, giving rise to the fundamental issue of non-stationarity: the environment is constantly changing due to the evolving behaviors of other agents from the perspective of any single agent. Recent works such as MAPPO (Yu et al., 2022), MADDPG (Lowe et al., 2017), and HAPPO/HATRPO (Kuba et al., 2021; Zhong et al., 2024) alleviate this challenge by using the *centralized training decentralized execution* (CTDE) framework, where agents assume access to global state information during training but rely on local information during execution. Although effective, the CTDE approach suffers from several drawbacks that limit its applicability in practical settings. First, it requires access to global information during training, which may be infeasible in large-scale systems due to communication bandwidth, latency, or privacy constraints —for example, wireless and ad-hoc networks often exhibit strong trade-offs between communication range, throughput, and latency (Seferagić et al., 2020), making long-range low-latency communication difficult to sustain. These limitations are particularly pronounced in off-road robotics, distributed sensing networks, and search-and-rescue settings, where cluttered terrain and unreliable links limit agents to short-range communication (Drew, 2021; Gielis et al., 2022). In such environments, relying on global information is impractical, motivating the need for learning frameworks that operate effectively with only local communication. Moreover, CTDE methods

often suffer from a train–test mismatch: agents are optimized with privileged global information that is unavailable during execution, which can lead to poor generalization once this information is removed. These limitations highlight the need for distributed MARL approaches that enable agents to learn cooperative strategies using only local observations and peer-to-peer communication among neighboring agents.

However, fully distributed learning techniques remain relatively underexplored compared to the centralized training approaches, partly due to the inherent complexity of the problem. Existing studies in this domain typically retain some form of centralization. For instance, Zhang et al. (2018) proposed a decentralized multi-agent actor–critic algorithms that use average consensus protocols (Tsitsiklis, 1984) to approximate global returns and value functions via neighbor communication, while actors update their policies independently. Although effective, this approach relies on simple averaging of value functions, which can yield suboptimal performance in heterogeneous teams with non-i.i.d. dynamics. Moreover, their framework assumes access to global state information for advantage estimation and thus cannot directly handle partial observability. In parallel, graph-based methods have been introduced to better capture structured communication among agents. Jiang et al. (2018) proposed graph convolution RL (DGN), where agents are represented as nodes in a dynamic graph and leverage Graph Attention Networks (GATs) (Veličković et al., 2017) to process node-level observations and actions. However, DGN shares both the Q-network and GAT parameters among all agents, preventing fully distributed training.

In addition to these decentralized methods, GNN augmented MARL approaches have also explored richer communication structures. For instance, the recent survey by Liu et al. (2024) outlines a broad class of GNN-based communication architectures (GNNComm-MARL) that enhance message routing, neighborhood selection, and multi-hop reasoning in cooperative tasks, but these methods continue to rely on CTDE and centralized critics. The attentional communication mechanism ATOC (Jiang & Lu, 2018) also adopts this paradigm: agents use a learned attention module to decide when to communicate; however, training still depends on a centralized critic and shared parameterization, which prevents fully distributed learning. Similarly, Goeckner et al. (2024) proposed a GNN-based patrolling framework where agents use deep message passing to overcome partial observability and communication disturbances; nevertheless, their actor–critic structure follows the CTDE paradigm, and training remains centralized. Another relevant line of work integrates information-theoretic objectives: Ding et al. (2023) introduced mutual-information–guided GNN communication to enhance representation quality in value-decomposition MARL. Despite its strong empirical performance, MARGIN requires centralized mixing of value functions and thus is not fully distributed. In the context of UAV coordination, Du et al. (2024) employed GNN observers to handle dynamic neighbor sets, and used transfer learning to accelerate QMIX-based training—again relying on centralized value mixing. Overall, existing GNN augmented MARL works demonstrate that graph-structured message passing improves coordination and robustness, especially under partial observability. However, none of these approaches enable fully distributed policy optimization, as they all depend on centralized critics, centralized value decomposition, or shared GNN parameterization across all agents. This leaves a significant gap: **how to design a MARL framework in which agents learn cooperative behaviors using only local observations, peer-to-peer communication, and fully distributed updates, without any reliance on centralized components or privileged information.**

We could bridge this gap by grounding MARL in distributed optimization techniques. For instance, decentralized stochastic gradient descent (D-SGD) (Lian et al., 2017; Assran et al., 2019) and classical distributed averaging protocols (Nedić & Ozdaglar, 2009; Tsitsiklis, 1984) provide strong theoretical foundations for consensus optimization over networks. Building on these insights, we introduce **Distributed Graph Attention Networks (D-GATs)**, which couple the expressiveness of GATv2 (Brody et al., 2021) with neighbor-averaged parameter sharing inspired by D-SGD. This design preserves dynamic, input-dependent attention while promoting consensus among agents in a fully distributed setting. While related works, such as GATTA (Tian et al., 2023), have applied graph attention to distributed supervised learning and personalization, it comes with higher computational overhead that scales poorly as the number of agents increases—making it less suitable for multi-agent reinforcement learning settings where efficiency is critical. In contrast, our Distributed Graph Attention Network (D-GAT) is designed specifically for MARL, focusing on global state inference through lightweight, input-dependent attention mechanisms that remain tractable even in large teams.

Building on D-GAT, we introduce **distributed graph-attention MAPPO (DG-MAPPO)**, a principled distributed MARL framework that removes the need for centralized training or global observability. DG-MAPPO integrates agents' local observations with global state inference from D-GAT and a shared/averaged team reward to learn collaborative policies that naturally scale to large teams. Unlike CTDE approaches, our fully distributed framework enables agents to infer global state during both training and execution, yielding more robust coordination at test time. We evaluate DG-MAPPO on the StarCraftII Multi-Agent Challenge (SMAC) (Samvelyan et al., 2019), Google Research Football Kurach et al. (2020), and Multi-Agent MuJoCo benchmarks for cooperative MARL, against strong CTDE baselines such as MAPPO, MAT-Dec (Wen et al., 2022), and HAPPO. Our experiments show that DG-MAPPO achieves consistently strong performance across diverse tasks, demonstrating its ability to handle both homogeneous and heterogeneous settings, scale to large teams, and learn effective collaboration without centralized training or privileged information.

Our contributions are threefold:

- We introduce **D-GAT**, a lightweight multi-hop communication module that enables agents to construct global state representations using only local message passing.
- We develop **DG-MAPPO**, a fully distributed MARL framework that learns cooperative policies solely from local observations, D-GAT–based state inference, and averaged team rewards—without any centralized training signal or privileged information.
- We provide extensive evidence on SMAC and Multi-Agent MuJoCo showing that DG-MAPPO matches or exceeds strong CTDE baselines, demonstrating that structured local communications alone support high-quality coordination even under sparse connectivity.

## 2 PRELIMINARIES

We begin by establishing the necessary background in this section. We begin by formulating the cooperative **multi-agent reinforcement learning** (MARL) problem as a **decentralized partially observable Markov decision process** (Dec-POMDP). We then describe how graph neural networks, particularly graph attention networks (GATs), can be utilized to model agent communication and representation learning. Finally, we review the policy gradient theorem in the multi-agent setting, which forms the foundation for our optimization framework. Throughout the paper, we denote matrices by bold uppercase letters (e.g., $\boldsymbol{X}$), vectors by bold lowercase letters (e.g., $\boldsymbol{x}$), local data with superscript $i$ (e.g., $x^i$), global data without superscript (e.g., $x$), and approximations with a hat (e.g., $\hat{x}$).

### 2.1 PROBLEM FORMULATION

We consider a distributed cooperative MARL problem formulated as a Dec-POMDP, represented by the tuple $\langle \mathcal{N}, \{\mathcal{O}^i\}_{i=1}^n, \{\mathcal{A}^i\}_{i=1}^n, R, P, \gamma \rangle$. Here, $\mathcal{N} = 1, \ldots, n$ is the set of agents. Each agent $i \in \mathcal{N}$ has an observation space $\mathcal{O}^i \subset \mathbb{R}^p$, where $p$ is the observation dimension, and an action space $\mathcal{A}^i \subset \mathbb{R}^q$, where $q$ is the action dimension. The joint observation and action spaces are $\mathcal{O} = \prod_{i=1}^n \mathcal{O}^i$ and $\mathcal{A} = \prod_{i=1}^n \mathcal{A}^i$, respectively. The transition kernel $P : \mathcal{O} \times \mathcal{A} \times \mathcal{O} \to [0, 1]$ defines the environment dynamics, $R : \mathcal{O} \times \mathcal{A} \to [-R_{\max}, R_{\max}]$ is the local reward function, and $\gamma \in [0, 1)$ is the discount factor.

**Remark 1** *Settings with local reward functions can be incorporated by computing a consensus-based average team reward (e.g., via average consensus protocol Saber & Murray (2003)).*

We model the multi-agent interaction structure as a dynamic graph $G = (\mathcal{N}, \mathcal{E})$, where nodes $(\mathcal{N})$ correspond to agents and edges $(\mathcal{E})$ denote available communication links which can change in real-time. At each time step $t$, agent $i$ receives a local observation $\boldsymbol{o}_t^i \in \mathcal{O}^i$ $\left(\boldsymbol{o}_t = [\boldsymbol{o}_t^1, \ldots, \boldsymbol{o}_t^n]^\top\right)$, communicates with nodes $j \in \mathcal{N}^i$, where $\mathcal{N}^i$ is some neighborhood of node $i$ (including $i$) in the graph $G$ over multiple-hops, and forms a local approximation of the global observation $\hat{\boldsymbol{o}}_t^i \in \hat{\mathcal{O}}^i$. Based on this, the agent selects an action $a_t^i$ from its policy $\pi^i$, which is the $i^{\text{th}}$ component of the joint policy $\pi = \prod_{i=1}^n \pi^i$. The transition kernel and the joint policy induce the marginal observation distribution $\rho_\pi(\cdot) = \sum_{t=0}^\infty \gamma^t Pr(\boldsymbol{o}_t \mid \pi)$ (Wen et al., 2022). All agents then receive an averaged team reward $R(\boldsymbol{o}_t, \boldsymbol{a}_t) = \frac{1}{N} \sum_{i \in \mathcal{N}} R^i(o^i, a^i)$ and observe $\boldsymbol{o}_{t+1}^i$.

We consider the fully cooperative setting in which all agents optimize a shared/averaged team reward. The goal is to learn local policies $\{\pi^i\}_{i=1}^n$ that maximize the expected discounted team return:

$$J\left(\pi\right) = \mathbb{E}_{\pi_\theta}\left[\sum_{t=0}^{\infty} \gamma^t R(o_t, a_t)\right] \tag{1}$$

## 2.2 Graph-Attention Networks

Graph neural networks (GNNs), such as GraphSAGE (Hamilton et al., 2017), learn node representations by aggregating information from local neighborhoods in a graph. At each layer, a node updates its embedding by combining its own features with those of its neighbors, typically using simple operations such as mean or sum. While effective, this uniform treatment of neighbors may fail to capture the varying importance of different connections.

GATs (Veličković et al., 2017; Brody et al., 2021) address this limitation by incorporating an attention mechanism into the aggregation process. Instead of assigning equal weight to all neighbors, GATs learn to adaptively highlight the most relevant nodes when computing new representations. Formally, for a node $i$ with feature vector $\boldsymbol{h}^i \in \mathbb{R}^d$ and neighborhood $\mathcal{N}^i$, GAT defines a shared attention function $e : \mathbb{R}^d \times \mathbb{R}^d \to \mathbb{R}$ to measure the importance of a neighbor $j$:

$$e(\boldsymbol{h}^i, \boldsymbol{h}^j) = \text{LeakyReLU}\left(\boldsymbol{q}^\top \left[\boldsymbol{W}\boldsymbol{h}^i \| \boldsymbol{W}\boldsymbol{h}^j\right]\right), \tag{2}$$

where $\boldsymbol{W} \in \mathbb{R}^{d' \times d}$ is a learnable linear transformation matrix, $\boldsymbol{q} \in \mathbb{R}^{2d'}$ is a trainable weight vector, and $\|$ denotes concatenation. The parameters $\boldsymbol{W}$ and $\boldsymbol{q}$ are shared across all nodes. These scores are normalized with a softmax across all neighbors:

$$\alpha_{ij} = \frac{\exp\left(e(\boldsymbol{h}^i, \boldsymbol{h}^j)\right)}{\sum_{j' \in \mathcal{N}^i} \exp(e(\boldsymbol{h}^i, \boldsymbol{h}^{j'}))}. \tag{3}$$

The attention coefficients $\alpha_{ij}$ encode the relative contribution of neighbor $j$ to node $i$. The updated representation of node $i$ is then computed as

$$\hat{\boldsymbol{h}}^i = \sigma\left(\sum_{j \in \mathcal{N}^i} \alpha_{ij} \boldsymbol{W}\boldsymbol{h}^j\right), \tag{4}$$

where $\sigma : \mathbb{R}^{d'} \to \mathbb{R}^{d'}$ is a nonlinear activation. By learning these attention weights, GATs provide a more flexible and expressive aggregation scheme than traditional GNNs, enabling the model to prioritize informative neighbors and downplay less relevant ones. In the MARL setting, this enables each agent to selectively integrate neighbor information when constructing a local approximation of the global state.

## 2.3 Policy Gradient Theorem for Multi-Agent Reinforcement Learning

Policy gradient methods provide a principled approach to optimizing parameterized policies by estimating the gradient of the expected return with respect to policy parameters. Extending this idea to the multi-agent setting raises unique challenges: agents act simultaneously, rewards are often observed only locally, and in the fully decentralized case, no central controller is available to aggregate global information.

Zhang et al. (2018) established a multi-agent policy gradient theorem for fully decentralized MARL under the assumption that the global states and actions are observable to all agents, while rewards remain local. This result forms the theoretical basis of their decentralized actor–critic algorithms.

**Theorem 1 (Policy Gradient for MARL (Zhang et al., 2018))** *Consider $N$ agents with local policies $\pi_{\theta^i}^i$ parameterized by $\theta^i$, and let the joint policy be $\pi_\theta = \prod_{i=1}^N \pi_{\theta^i}^i, \theta = [\theta^1, \ldots, \theta^n]^\top$. The collective objective is to maximize the globally averaged return $J(\pi_\theta)$ defined in Equation (1). Then, for each agent $i$, the policy gradient with respect to $\theta^i$ is given by*

$$\nabla_{\theta^i} J(\pi_\theta) = \mathbb{E}_{\boldsymbol{o} \sim \rho_{\pi_\theta}, \boldsymbol{a} \sim \pi_\theta}\left[\nabla_{\theta^i} \log \pi_{\theta^i}^i(\boldsymbol{o}_t, \boldsymbol{a}_t^i) A_\theta(\boldsymbol{o}_t, \boldsymbol{a}_t)\right], \tag{5}$$

where $A_\theta(\boldsymbol{o}_t, \boldsymbol{a}_t) = R(\boldsymbol{o}_t, \boldsymbol{a}_t) + \gamma V_\phi(\boldsymbol{o}_{t+1}) - V_\phi(\boldsymbol{o}_t)$ *is the global advantage function, and* $V_\phi :$ $\mathcal{O} \to \mathbb{R}$ *is the global state-value function parameterized by* $\phi$.

This theorem shows that each agent can compute its policy gradient update using only its local policy parameters and an estimate of the global advantage function. Theorem 1 provides the theoretical foundation for distributed MARL, upon which we build our distributed MARL algorithm in the next section.

## 3 METHOD

In this section, we present DG-MAPPO, a distributed MARL algorithm that is both simple and scalable. Unlike the widely adopted CTDE paradigm, our approach does not assume access to global state information during either training or execution. Instead, coordination emerges organically through multi-hop message passing over a connected communication graph.

**Assumption 1 (Connected Communication Graph)** *The communication graph $G$ is* connected; *that is, for any two distinct agents $i \neq j$, there exists at least one path from $i$ to $j$ in $G$.*

This assumption significantly relaxes the stronger requirement of centralized information sharing commonly made in prior MARL frameworks. Building on this, we formalize the notion of distributed MARL as follows:

**Definition 1 (Distributed MARL)** *Consider a system of $n$ agents operating on a connected graph $G$. Each agent $i$ observes only its local information $\boldsymbol{o}^i$ and can communicate it with its neighbors. A distributed MARL algorithm requires each agent to learn both its policy and value function using solely its local observations and peer-to-peer communication, without relying on centralized training or access to the global state.*

This definition distinguishes distributed approaches from purely decentralized ones: the former leverage communication among agents to enable collaboration, whereas the latter operate independently without inter-agent communication (e.g., the decentralized execution in CTDE). In what follows, we first introduce D-GAT, our communication module that enables global state inference via multi-hop message passing in a fully distributed manner. We then introduce our DG-MAPPO algorithm which learns collaborative policies entirely from local observations and peer-to-peer communication.

### 3.1 DISTRIBUTED GRAPH ATTENTION NETWORKS

GATs (Veličković et al., 2017) are a powerful tool for learning from graph-structured data, but their standard formulation relies on globally shared attention parameters, preventing deployment in fully distributed settings. We address this by introducing D-GAT, where each agent independently maintains and updates its own local attention parameters. This design ensures that message aggregation remains attention-driven while fully respecting the real-time communication constraints. In addition, we adopt the dynamic attention formulation of GATv2 (Brody et al., 2021), which extends the original GAT by enabling input-dependent query–key interactions, thereby enhancing representational expressiveness. The overall framework of D-GAT is illustrated in Figure 1.

A single D-GAT layer for node $i$ operates as follows. For a node $i$ with feature vector $\boldsymbol{h}^i \in \mathbb{R}^d$ and neighborhood $\mathcal{N}^i$, we define a local attention function $e^i : \mathbb{R}^d \times \mathbb{R}^d \to \mathbb{R}$ that measures the importance of neighbor $j$ as:

$$e^i(\boldsymbol{h}^i, \boldsymbol{h}^j) = {\boldsymbol{q}^i}^\top \text{LeakyReLU}\left(\boldsymbol{W}^i[\boldsymbol{h}^i\|\boldsymbol{h}^j]\right). \tag{6}$$

where $\boldsymbol{W}^i \in \mathbb{R}^{d' \times 2d}$ is a learnable linear projection of agent $i$, $\boldsymbol{q}^i \in \mathbb{R}^{d'}$ is a trainable weight vector of agent $i$, and $\|$ is the concatenation operator. We then perform score normalization and feature aggregation as:

$$\alpha_{ij}^i = \frac{\exp\left(e^i(\boldsymbol{h}^i, \boldsymbol{h}^j)\right)}{\sum_{j' \in \mathcal{N}^i} \exp(e^i(\boldsymbol{h}^i, \boldsymbol{h}^{j'}))}, \quad \hat{\boldsymbol{h}}^i = \sigma\left(\sum_{j \in \mathcal{N}^i} \alpha_{ij}^i \boldsymbol{h}^j\right), \tag{7}$$

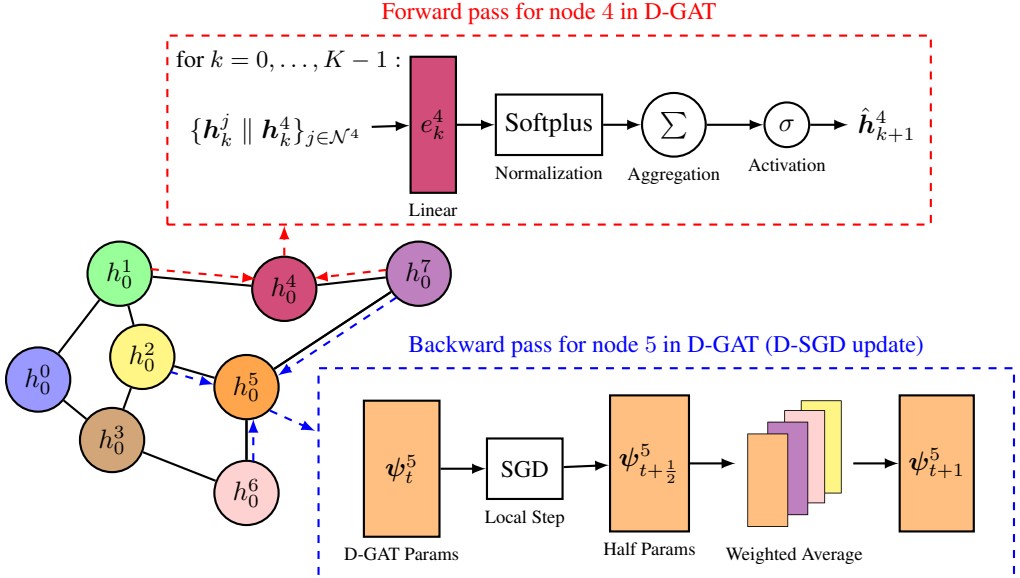

Figure 1: Illustration of the forward and backward passes in the proposed D-GAT framework. **1) Forward pass (top, red):** At each layer $k$, node 4 aggregates information from its neighbors $\{h_k^j \| h_k^4\}_{j \in \mathcal{N}^4}$ by applying a linear transformation, Softplus normalization, summation-based aggregation, and a nonlinearity $\sigma$ to produce the updated embedding $\hat{h}_{k+1}^4$. This process is repeated for $k = 0, \ldots, K-1$, where $K$ is the predefined number of hops. **2) Backward pass (bottom, blue):** Node 5 updates its local D-GAT parameters $\psi^5$ via decentralized stochastic gradient descent (D-SGD). First, a local step computes the half-step parameters $\psi_{t+\frac{1}{2}}^5$ using the local update step of Equation (8). Next, a neighbor averaging step mixes parameters from neighbors via the neighbor averaging step of Equation (8) to get the updated D-GAT parameters $\psi_t^5$, enabling distributed training without a central coordinator.

where $\hat{h}^i$ is the updated vector representation of agent $i$ computed as an attention-weighted aggregation of its neighbors' features, followed by a nonlinear activation $\sigma(\cdot)$. We stack $n$ such layers (equal to the number of agents) to facilitate multi-hop communication, ensuring that every agent $i \in \mathcal{N}$ can exchange information with all other agents $j \in \mathcal{N}$.

The distributed design of D-GAT introduces a fundamental challenge in MARL: each agent updates its local attention parameters solely to maximize its own performance. Such locally selfish updates can impede the formation of a coherent global representation, which is essential for effective coordination. Consequently, agents may struggle to approximate the global state, leading to suboptimal joint performance. To mitigate this issue, we propose a two-step solution. Firstly, inspired by decentralized stochastic gradient descent (D-SGD) (Lian et al., 2017), we update each agent's graph network parameters via local SGD and then average them with its immediate neighbors. Mathematically, it is given as:

$$
\begin{aligned}
\textbf{(Local step)} \quad & \psi_{t+\frac{1}{2}}^i = \psi_t^i - \eta_t \, \widehat{\nabla} \ell^i \big( \psi_t^i; \xi_t^i \big), \\
\textbf{(Neighbor averaging)} \quad & \psi_{t+1}^i = \sum_{j \in \mathcal{N}^i} c(i,j) \, \psi_{t+\frac{1}{2}}^j, \qquad i = 1, \ldots, n,
\end{aligned} \tag{8}
$$

where $\psi^i = \{W^i, q^i\}$ are the local graph attention network parameters, $\widehat{\nabla} \ell^i \big( \psi_t^i; \xi_t^i \big)$ is a stochastic gradient computed from local data/minibatch $\xi_t^i$, $\eta_t$ is the learning step-size, and $c(i,j)$ is the consensus weight between agent $i$ and $j$ consistent with the communication graph given by,

$$
c(i,j) = \begin{cases} \dfrac{1}{|\mathcal{N}^i|}, & \text{if } j \in \mathcal{N}^i, \\ 0, & \text{otherwise.} \end{cases} \tag{9}
$$

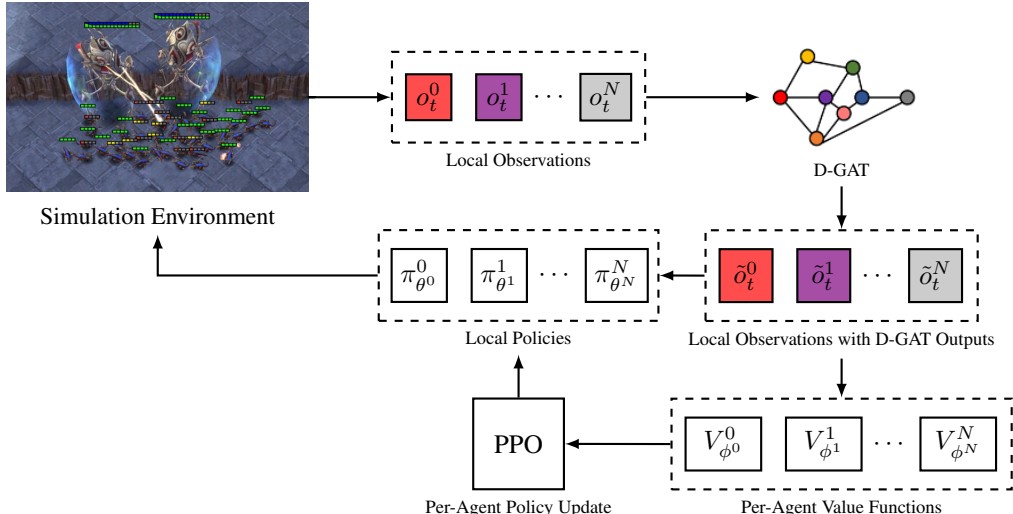

Figure 2: **DG-MAPPO Framework.** Each agent receives raw local observations $\boldsymbol{o}_t^i$ from the environment. Agents then communicate with neighbors using D-GAT to get global state inference $\hat{\boldsymbol{o}}_t^i$ which is then concatenated with raw local observations to get $\tilde{\boldsymbol{o}}_t^i = [\boldsymbol{o}_t^i, \hat{\boldsymbol{o}}_t^i]$. This combined observation is used by both local policies $\pi_{\theta^i}^i$ to generate actions and by value functions $V_{\phi^i}^i$ to estimate global returns. PPO performs per-agent policy updates using the advantage estimates derived from GAE.

Here, $|\mathcal{N}^i|$ is the degree of node $i$. This D-SGD procedure is equivalent to an average consensus step over the local D-GAT parameters, ensuring that agents gradually align their representations with those of their neighbors. By doing so, the network parameters are regularized across the graph, which improves the generalizability of the learned representations. Importantly, since the local updates still follow the GATv2 architecture, agents can compute dynamic, input-dependent attention weights as in Brody et al. (2021), thereby preserving the expressive power of attention mechanisms while operating in a distributed setting.

Moreover, to facilitate global state inference, we introduce a consensus regularization objective for D-GAT that explicitly encourages neighboring agents to align their learned representations. Concretely, in addition to the value (Equation (13)) and policy losses (Equation (14)), each agent also minimizes a consensus loss with respect to its neighbors, given by

$$\mathcal{L}_{\text{consensus}}^i(\boldsymbol{\psi}^i) = \alpha_{\text{consensus}} \frac{1}{|\mathcal{N}^i|} \sum_{j \in \mathcal{N}^i} \text{MSE}(\hat{\boldsymbol{h}}_K^i, \hat{\boldsymbol{h}}_K^j), \tag{10}$$

where $\alpha_{\text{consensus}}$ controls the strength of the regularization, $\hat{\boldsymbol{h}}_K$ is the output of D-GAT (with $K$ hops) forward pass, and $\text{MSE}(\cdot, \cdot)$ denotes the mean-squared error. Intuitively, D-GAT integrates D-SGD with a consensus regularization objective, providing a communication framework that enables agents to extract essential global state information while suppressing irrelevant information from less important neighbors. For instance, D-GAT enables agents to prioritize information from collaborators with strong influence on their outcomes, while downweighting signals from agents whose actions have little impact.

## 3.2 DISTRIBUTED GRAPH-ATTENTION MAPPO

We now introduce our distributed MARL framework, **DG-MAPPO**, illustrated in Figure 2. The central idea is that, given access to global state information and a globally shared (or averaged) reward, each agent $i \in \mathcal{N}$ can independently learn a global state value function $V_{\phi^i}^i : \mathcal{O} \to \mathbb{R}$ defined as

$$V_{\phi^i}^i(\boldsymbol{o}_t) = \mathbb{E}_{a_t \sim \pi_\theta} \left[ \sum_{t=0}^T \gamma^t R(\boldsymbol{o}_t, \boldsymbol{a}_t) \,\middle|\, \boldsymbol{o}_0 = \boldsymbol{o}_t \right]. \tag{11}$$

In practice, however, agents in our distributed setting cannot directly observe the true global state. To overcome this limitation, we incorporate the **D-GAT communication module** (see Section 3.1), which enables agents to perform multi-hop message passing after each local observation. Through this process, agents obtain an informative approximation of the global state, denoted as $\hat{\boldsymbol{o}}_t^i \in \hat{\mathcal{O}}^i$.

Training the value function solely on $\hat{\boldsymbol{o}}_t^i$ can lead to high variance, particularly in the early stages of learning, which risks destabilizing the training process. To address this, we provide each agent with a concatenated input combining its own local observation and the global state approximation:

$$\tilde{\boldsymbol{o}}_t^i = [\boldsymbol{o}_t^i \parallel \hat{\boldsymbol{o}}_t^i] \in \tilde{\mathcal{O}}^i \tag{12}$$

This representation ensures that agents retain a reliable self-signal while progressively benefiting from improved global context. Each agent's critic is then trained by minimizing the Bellman error:

$$\mathcal{L}_{\text{critic}}^i(\phi^i) = \mathbb{E}_{a_t \sim \pi_\theta} \left[ \sum_{t=0}^{T-1} R(\boldsymbol{o}_t, \boldsymbol{a}_t) + \gamma V_{\phi^i}^i(\tilde{\boldsymbol{o}}_{t+1}^i) - V_{\phi^i}^i(\tilde{\boldsymbol{o}}_t^i) \right]^2 . \tag{13}$$

Although agents cannot directly access the joint global policy $\pi_\theta$, they can still learn state-value functions consistent with it. This is possible because agents experience a common reward signal—either provided by a global reward mechanism or obtained through averaging local rewards via consensus, and they condition the state-value function on $\tilde{\boldsymbol{o}}_t^i$, consisting of the global state representation. Agents can now estimate the global advantage function $A_\theta^i : \tilde{\mathcal{O}}^i \times \mathcal{A} \to \mathbb{R}$ leveraging the shared/average reward $R(\boldsymbol{o}_t, \boldsymbol{a}_t)$ and the local estimate of the global value function $V^i$. In practice, the agents use **generalized advantage estimation** (GAE) (Schulman et al., 2015) to independently approximate a low-bias, low-variance global advantage estimate leveraging the local stored trajectories.

Following Theorem 1, the policy parameters of each agent $\theta^i$ are updated using a clipped policy gradient objective, as in PPO (Schulman et al., 2017),

$$\mathcal{L}_{\text{DG-MAPPO}}^i(\theta^i) = \mathbb{E}_t \left[ \min \left( \frac{\pi_{\theta^i}^i(a_t^i \mid \tilde{\boldsymbol{o}}_t^i)}{\pi_{\theta_{\text{old}}^i}^i(a_t^i \mid \tilde{\boldsymbol{o}}_t^i)}, \text{ clip} \left( \frac{\pi_{\theta^i}^i(a_t^i \mid \tilde{\boldsymbol{o}}_t^i)}{\pi_{\theta_{\text{old}}^i}^i(a_t^i \mid \tilde{\boldsymbol{o}}_t^i)}, 1 \pm \epsilon \right) \right) A_\theta^i(\tilde{\boldsymbol{o}}_t^i, \boldsymbol{a}_t) \right],$$

where $\epsilon$ is the clip parameter. A brief derivation of the DG-MAPPO policy gradient loss is provided in Appendix A.3. Overall, using DG-MAPPO, each agent performs local actor–critic updates using only its own observation and a local approximation of the global state, acquired through multi-hop communication, while coordination emerges organically from the shared reward structure and multi-hop message passing. The pseudocode is provided in the Appendix A.2. A comprehensive analysis of communication overhead, and cost analysis is provided in Appendix A.4 A.5.

## 4 RESULTS

Our distributed MARL framework offers a principled alternative to the widely adopted CTDE paradigm for cooperative MARL. Instead of relying on centralized critics and global information, our approach enables agents to collaborate using only local observations and peer-to-peer communication. By leveraging the dynamic communication graph, agents can mimic—and often surpass—the benefits of CTDE methods. A distinctive advantage is that communication is actively used during both training and execution, allowing agents to maintain awareness of their neighbors' states and adapt their coordination in real time.

We evaluate DG-MAPPO on StarCraftII Multi-Agent Challenge (SMAC), Google Research Football (GFootball) and Multi-Agent MuJoCo benchmarks, where CTDE approaches have shown SOTA performance. We adopt the strongest reported results for SMAC from the existing literature without re-running the baseline algorithms. For MA-MuJoCo, we follow the original implementations and parameter settings to reproduce each method's best-performing configuration.

### 4.1 EXPERIMENT SETUP

While CTDE baselines leverage global observations available in SMAC, GFootball, and Multi-Agent MuJoCo, DG-MAPPO operates strictly from local observations. At each timestep, agents

Table 1: Performance evaluations of win rate and standard deviation on the SMAC benchmark for default sight range value "9".

| Task | Difficulty | MAT-Dec | MAPPO | HAPPO | DG-MAPPO |
|---|---|---|---|---|---|
| 3m | Easy | **100.0**$_{(1.1)}$ | **100.0**$_{(0.4)}$ | **100.0**$_{(1.2)}$ | **100.0**$_{(1.4)}$ |
| 8m | Easy | 97.2$_{(2.5)}$ | 96.8$_{(2.9)}$ | 97.5$_{(1.1)}$ | **100.0**$_{(1.4)}$ |
| MMM | Easy | 98.1$_{(2.1)}$ | 95.6$_{(4.5)}$ | 81.2$_{(22.9)}$ | **100.0**$_{(1.6)}$ |
| 5m vs 6m | Hard | 83.1$_{(4.6)}$ | 88.2$_{(6.2)}$ | 77.5$_{(7.2)}$ | **88.7**$_{(4.7)}$ |
| 8m vs 9m | Hard | 95.0$_{(4.6)}$ | 93.8$_{(3.5)}$ | 86.2$_{(4.4)}$ | **95.0**$_{(4.1)}$ |
| 10m vs 11m | Hard | **100.0**$_{(2.0)}$ | 96.3$_{(5.8)}$ | 87.5$_{(6.7)}$ | **100.0**$_{(1.4)}$ |
| 25m | Hard | 86.9$_{(5.6)}$ | **100.0**$_{(2.7)}$ | 95.0$_{(2.0)}$ | 95.3$_{(3.1)}$ |
| MMM2 | Hard+ | 91.2$_{(5.3)}$ | 81.8$_{(10.1)}$ | 88.8$_{(2.0)}$ | **98.9**$_{(1.2)}$ |
| 6h vs 8z | Hard+ | 93.8$_{(4.7)}$ | 88.4$_{(5.7)}$ | 76.2$_{(3.1)}$ | **95.0**$_{(2.7)}$ |
| 3s5z vs 3s6z | Hard+ | 85.3$_{(7.5)}$ | 84.3$_{(19.4)}$ | 82.8$_{(21.2)}$ | **91.9**$_{(10.7)}$ |

Table 2: Performance evaluations of win rate and standard deviation on the SMAC benchmark for clipped sight range value "4" across different hop values.

| Task | Num-Agents | Difficulty | 1-Hop | $\frac{N}{2}$-Hops | $N$-Hops | Steps |
|---|---|---|---|---|---|---|
| 6h vs 8z | 6 | Hard+ | **77.08**$_{(7.6)}$ | **83.68**$_{(10.0)}$ | **83.75**$_{(7.7)}$ | 4e7 |
| MMM2 | 10 | Hard+ | **90.62**$_{(3.1)}$ | **92.7**$_{(3.6)}$ | **93.1**$_{(2.6)}$ | 4e7 |

communicate through the D-GAT module to construct an inferred global representation, which is then used for action selection alongside the shared environment reward. To preserve fully distributed training, each agent maintains its own local dataset and performs updates independently. Parameter averaging with local neighbors is applied only to the D-GAT networks, as described in Section 3.1. Since the communication topology in SMAC and GFootball evolves over time, we record the average node degree at the end of each episode (Appendix A.9) to characterize graph connectivity. In contrast, we define a sparse fixed communication topology for the Multi-Agent MuJoCo environment, where communication is restricted to physically adjacent agents (joints).

### 4.2 Performance on Cooperative Benchmarks

Table 1 compares DG-MAPPO with strong CTDE baselines (MAPPO, HAPPO, and MAT-Dec) on the SMAC benchmark with a default communication range of 9 *units* for each agent. DG-MAPPO achieves consistently strong results across diverse tasks, ranging from small homogeneous battles to challenging heterogeneous and large-scale scenarios. Notably, the *25m* scenario highlights DG-MAPPO's ability to scale to larger teams even under a sparse communication topology (see Appendix A.9). **To the best of our knowledge, this is the first distributed MARL approach to match CTDE-level performance in teams of up to 25 agents.** To further assess DG-MAPPO's performance in highly sparse settings, we reduce the communication range to 4 units and evaluate the method on two "Hard+" scenarios, *6h vs 8z* and *MMM2*. The corresponding evaluation win rates across different hop values are reported in Table 2, with performance and average node-degree curves provided in Appendix A.6.

These results show that DG-MAPPO not only matches but in several cases surpasses strong CTDE baselines when agents operate with relatively dense communication (Table 1). More importantly, DG-MAPPO maintains competitive performance even when the communication network is made highly sparse, as demonstrated in the clipped-range experiments for *6h vs 8z* and *MMM2* (Table 2). A notable trend across both settings is that DG-MAPPO learns effectively even with a small number of hops—often achieving near-optimal win rates with $K = N/2$ or even $K = 1$—thereby reducing communication and computation overhead with only marginal performance degradation. Figure 3a shows similar performance trend of DG-MAPPO compared to CTDE baselines in the GFootball environment.

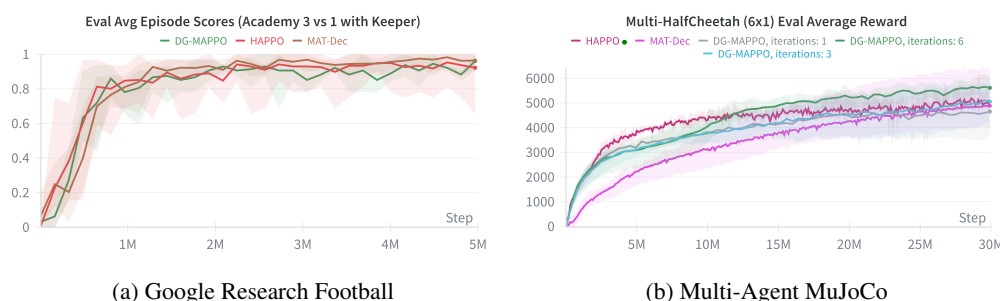

(a) Google Research Football                    (b) Multi-Agent MuJoCo

Figure 3: Evaluation performance of DG-MAPPO compared to the CTDE baselines. (a) Results on the Google Research Football *Academy 3 vs 1 with Keeper* scenario. (b) Results on the Multi-Agent MuJoCo *Multi-HalfCheetah (6×1)* task.

The results on Multi-Agent MuJoCo (Figure 3b) further highlight DG-MAPPO's robustness in continuous-control settings. Unlike SMAC, these tasks provide dense proprioceptive observations and enforce a fixed communication topology in which each agent exchanges information only with its physically adjacent joints. Even under these constraints, DG-MAPPO—operating solely on local observations and learned multi-hop message passing—achieves returns on the $6{\times}1$ Multi-HalfCheetah benchmark comparable to those of CTDE baselines. Increasing the hop count from $K = 1$ to $K = 3$ yields improved sample efficiency, closely tracking CTDE training curves, while $K = 6$ surpasses the CTDE baseline. This mirrors our findings in SMAC: only a small number of message-passing hops are needed to match CTDE performance, and additional hops offer incremental gains at the cost of higher communication and computation. Overall, these results demonstrate that DG-MAPPO scales naturally to continuous-control settings and maintains strong performance under restricted communication, reinforcing its applicability beyond discrete cooperative tasks.

Please refer to Appendix A.7 for a detailed analysis of the impact of attention-based aggregation, hop count, and the Consensus Loss regularization.

## 5   CONCLUSION

We presented DG-MAPPO, a fully distributed MARL framework that leverages multi-hop message passing through D-GAT to learn collaborative policies without any centralized controller or privileged observations. Across SMAC, GFootball, and Multi-Agent MuJoCo environments, DG-MAPPO consistently achieves performance on par with, and often exceeding, strong CTDE baselines—despite operating under significantly more restrictive information conditions. These results demonstrate that structured local communication, when combined with expressive graph-based aggregation, is sufficient to enable high-quality cooperative behavior in complex partially observable environments. Our findings also shed light on the practical robustness of distributed communication. DG-MAPPO performs reliably across diverse settings, exhibits stable training dynamics, and scales naturally across both discrete and continuous control domains. Notably, the algorithm maintains strong performance even when restricted to sparse communication networks, highlighting its resilience to limited communication depth and its suitability for environments where long-range information flow is inherently constrained. In addition, DG-MAPPO maintains competitive performance in larger team scenarios, such as the *25m*, demonstrating that distributed communication alone can effectively support long-range coordination in challenging multi-agent systems. Overall, DG-MAPPO represents an important step toward scalable, decentralized, and deployment-ready multi-agent learning. By demonstrating that competitive performance can be achieved without relying on centralized training assumptions, our work broadens the path toward more resilient, realistic, and scalable MARL systems capable of operating in dynamic and uncertain real-world environments.

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

# A APPENDIX

## A.1 USE OF LLMS

We used a large language model (OpenAI's ChatGPT) as a writing assistant to help polish the clarity, grammar, and readability of certain sections of the paper (e.g., abstract, introduction, and conclusion). The model was not used for generating research ideas, designing experiments, or analyzing results. All technical content, experiments, and conclusions were conceived and validated solely by the authors.

## A.2 PSEUDO CODE FOR DG-MAPPO

---
**Algorithm 1** Distributed Graph-Transformer MAPPO

---
**Input:** Number of agents and hops $n$, learning rate $\alpha$, episodes $K$, steps per episode $T$
**Initialize:** D-GAT $\{\psi^i\}_{i \in \mathcal{N}}$, Critic $\{\phi^i\}_{i \in \mathcal{N}}$, Policy $\{\theta^i\}_{i \in \mathcal{N}}$, Replay Buffer $\{\boldsymbol{\xi}^i\}_{i \in \mathcal{N}}$
**for** $k = 0, 1, \ldots, K - 1$ **do**
  **for** $t = 0, 1, \ldots, T - 1$ **do**
    Receive local observations $\{\boldsymbol{o}_t^i\}_{i \in \mathcal{N}}$ from environment.
    Perform multi-hop communication using D-GAT to infer global state $\{\hat{\boldsymbol{o}}_t^i\}_{i \in \mathcal{N}}$.
    Sample actions using local policies $a_t^i \sim \pi_{\theta^i}^i \ \ \forall i \in \mathcal{N}$.
    Perform the joint action $\boldsymbol{a}_t$ in the environment and observe joint reward $R(\boldsymbol{o}_t, \boldsymbol{a}_t)$.
    Store $(\boldsymbol{o}_t^i, \hat{\boldsymbol{o}}_t^i, \boldsymbol{a}_t^i, R(\boldsymbol{o}_t, \boldsymbol{a}_t))$ in the buffer $\{\boldsymbol{\xi}^i\}$
  **end for**
  Sample random minibatch $\xi^i$ from $\boldsymbol{\xi}^i$
  Infer state-values for all agents $\{V_{\phi^i}^i(\tilde{\boldsymbol{o}}^i)\}_{i \in \mathcal{N}}$, where $\tilde{\boldsymbol{o}}^i = (\boldsymbol{o}^i \| \hat{\boldsymbol{o}}^i)$.
  Calculate $\{\mathcal{L}_{\text{critic}}^i(\phi^i)\}_{i \in \mathcal{N}}$ using Equation (13).
  Estimate global advantage $A^i$ based on $V_{\phi^i}^i(\tilde{\boldsymbol{o}}^i)$ and $R(\boldsymbol{o}_t, \boldsymbol{a}_t) \ \ \forall i \in \mathcal{N}$ using GAE.
  Compute policy loss $\{\mathcal{L}_{\text{PPO}}^i(\theta^i)\}_{i \in \mathcal{N}(\phi^i)}$ using Equation (14).
  Compute D-GAT consensus regularizer loss $\{\mathcal{L}_{\text{consensus}}^i(\psi^i)\}_{i \in \mathcal{N}}$ using Equation (10).
  Update D-GAT, value critic, and policy networks by minimizing $\mathcal{L}_{\text{critic}}^i(\phi^i) + \mathcal{L}_{\text{PPO}}^i(\theta^i) + \mathcal{L}_{\text{consensus}}^i(\psi^i) \ \ \forall i \in \mathcal{N}$ using gradient decent.
**end for**

---

## A.3 DERIVATION OF THE DG-MAPPO POLICY LOSS FROM THE MARL POLICY GRADIENT THEOREM

We provide a brief derivation showing how the decentralized MARL policy gradient theorem (Theorem 1) leads directly to the PPO surrogate loss used in DG-MAPPO.

The Multi-agent policy gradient theorem states that the policy gradient for each agent $i$ can be computed with respect to local parameters $\theta^i$ using Eq. 5, as long as we can access a global advantage function $A_\theta(\boldsymbol{o}_t, \boldsymbol{a}_t)$.

In DG-MAPPO, the agent acts on the augmented local observation $\tilde{\boldsymbol{o}}^i = [o^i \| \hat{o}^i]$, and the global advantage is approximated locally using the shared or averaged reward $R(\boldsymbol{o}, \boldsymbol{a})$ and the local critic $V_{\phi^i}^i(\tilde{o}^i)$ as,

$$A_\theta^i(\tilde{\boldsymbol{o}}_t^i, \boldsymbol{a}_t) = R(\boldsymbol{o}_t, \boldsymbol{a}_t) + \gamma V_{\phi^i}^i(\tilde{\boldsymbol{o}}_{t+1}^i) - V_{\phi^i}^i(\tilde{\boldsymbol{o}}_t^i) \tag{14}$$

Thus,

$$\nabla_{\theta^i} J(\pi_\theta) \approx \mathbb{E}\Big[\nabla_{\theta^i} \log \pi_{\theta^i}^i(a^i \mid \tilde{o}^i) A_\theta^i(\tilde{\boldsymbol{o}}_t^i, \boldsymbol{a}_t)\Big]. \tag{15}$$

Trajectories are collected under the old policy $\pi_{\theta_{\text{old}}^i}^i$. Rewriting equation 15 using importance sampling yields

$$\nabla_{\theta^i} J(\pi_\theta) \approx \mathbb{E}_{\pi_{\theta_{\text{old}}^i}^i}\big[r^i(\theta^i) \nabla_{\theta^i} \log \pi_{\theta^i}^i(a_i \mid \tilde{o}^i) A_\theta^i(\tilde{\boldsymbol{o}}_t^i, \boldsymbol{a}_t)\big], \tag{16}$$

where $r^i(\theta^i) = \frac{\pi_{\theta^i}^i(a^i | \tilde{o}^i)}{\pi_{\theta_{\text{old}}^i}^i(a^i | \tilde{o}^i)}$ is the probability ratio. Using $\nabla_{\theta^i} \log \pi = \frac{1}{r^i} \nabla_{\theta^i} r^i$, this corresponds to maximizing the standard surrogate loss

$$\mathcal{L}_{\text{PG}}^i(\theta^i) = \mathbb{E}\big[r^i(\theta_i) A^i\big]. \tag{17}$$

Large deviations of $r^i(\theta^i)$ can destabilize decentralized learning; following the approach of PPO, we therefore replace the unconstrained surrogate equation 17 with the clipped surrogate loss

$$\mathcal{L}_{\text{DG-MAPPO}}^i(\theta^i) = \mathbb{E}\big[\min\big(r^i(\theta^i)A^i,\ \text{clip}(r^i(\theta^i), 1 - \epsilon, 1 + \epsilon) A^i\big)\big], \tag{18}$$

which preserves the ascent direction of the policy gradient while preventing excessively large updates. This is the policy loss optimized by each agent in DG-MAPPO.

### A.4 COMMUNICATION COMPLEXITY AND OVERHEAD ANALYSIS

To demonstrate the scalability of DG-MAPPO, we compare its communication requirements with those of standard CTDE baselines (e.g., MAPPO, MAT-Dec). We distinguish between the *training phase* (gradient and parameter synchronization) and the *execution phase* (inference and action selection), and explicitly account for how message size, hop count, and network structure enter the communication complexity. An overview is provided in Table 3.

#### A.4.1 TRAINING PHASE

Standard CTDE methods rely on a central learner that aggregates observations and actions from all $N$ agents to compute joint value functions or policy updates. Let $|O^i|$ and $|A^i|$ denote the dimensions of agent $i$'s observation and action spaces, respectively. The total amount of data sent to the central learner per update can be written as

$$\mathcal{C}_{\text{CTDE}}^{\text{train}} = \sum_{i=1}^{N} \big(|O^i| + |A^i|\big), \tag{19}$$

which scales as $\mathcal{O}\big(N(|O| + |A|)\big)$ under homogeneous agents. When the communication range is physically constrained, distant agents must route their data through multi-hop paths to reach the central server, resulting in increased latency and physical communication costs within the network.

In contrast, DG-MAPPO eliminates the central sink. Each agent $i$ maintains its own local policy parameters $\theta^i$ and critic parameters $\phi^i$, and performs fully local actor–critic updates. Communication during training is restricted to:

1. **Message passing**: exchanging feature embeddings $h^i \in \mathbb{R}^{|h|}$ with immediate neighbors $\mathcal{N}^i$.

2. **Representation consensus**: averaging D-GAT parameters $\psi^i$ with neighbors $\mathcal{N}^i$ using the D-SGD update.

3. **Reward consensus**: exchanging local reward values for average consensus when a globally shared reward is unavailable.

At each hop, agent $i$ transmits a single embedding vector of dimension $|h|$ to every neighbor $j \in N^i$. The per-hop message-passing cost over the whole network is therefore

$$\mathcal{C}_{\text{DG}}^{\text{MP, 1-hop}} = \sum_{i=1}^{N} |N^i| \, |h|. \tag{20}$$

For $K$ hops, the communication cost during training due to D-GAT message passing is

$$\mathcal{C}_{\text{DG}}^{\text{MP}} = K \sum_{i=1}^{N} |N^i| \, |h|. \tag{21}$$

In addition, D-GAT parameter averaging incurs a cost

$$\mathcal{C}_{\text{DG}}^{\text{param}} = \sum_{i=1}^{N} |N^i| \, |\psi|, \tag{22}$$

where $|\psi|$ denotes the number of parameters in the local D-GAT instance.

Combining these contributions, the total training-time communication cost of DG-MAPPO can be written as

$$\mathcal{C}_{\text{DG}}^{\text{train}} = \mathcal{C}_{\text{DG}}^{\text{MP}} + \mathcal{C}_{\text{DG}}^{\text{param}}. \tag{23}$$

Comparing Eq. 24 and Eq. 23, it is clear that CTDE methods have lower communication cost compared to our fully distributed MARL method in ideal settings. However, when the communication is restricted, CTDE methods will also have to route their information to the central controller via multi-hop communication.

$$\mathcal{C}_{\text{CTDE-MultiHop}}^{\text{train}} = \sum_{i=1}^{N} K^i \big(|O^i| + |A^i|\big), \tag{24}$$

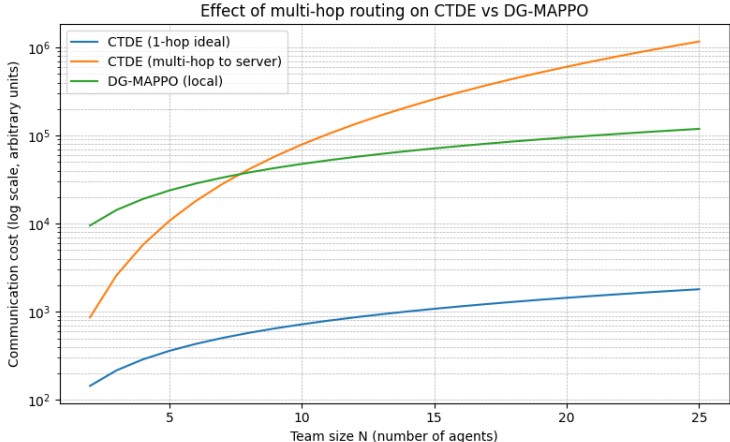

Figure 4: Comparison of communication cost for CTDE and DG-MAPPO as team size N increases. All methods assume identical per-message and per-observation sizes. The DG-MAPPO curve uses a hop budget of N/2, consistent with our ablation findings on effective message-passing depth. The multi-hop CTDE baseline models a worst-case 1-D network topology. Results are shown on a log scale to highlight differences in growth rates.

Table 3: Communication and Computational Complexity Comparison. $N$ denotes the number of agents, $|\mathcal{O}|$ and $|\mathcal{A}|$ the observation and action space sizes, $\psi$ the D-GAT parameters, and $K$ the number of communication hops.

| Feature | CTDE (e.g., MAPPO) | DG-MAPPO (Ours) |
|---|---|---|
| **Training Architecture** | Centralized Server / Coordinator | Fully Distributed (Peer-to-Peer) |
| **Training Comm. Pattern** | Many-to-One (Gather + Broadcast) | Neighbor-to-Neighbor (Mesh) |
| **Comm. Bottleneck** | Central Server Bandwidth ($\mathcal{O}(N)$) | Uniform Link Load ($\mathcal{O}(|\mathcal{N}^i|)$) |
| **Data Transferred (Train)** | Obs., Actions, Global State, Gradients | D-GAT Params ($\psi$), Feature Vectors ($h^i$) |
| **Execution Comm.** | None (Silent) | Feature Vectors ($h^i$) via D-GAT |
| **Dependency on Range** | High (Requires global reach) | Low (Localized to radius $R$) |
| **Scalability** | Limited by Central Server I/O | Linearly Scalable with Team Size |

which scales as $\mathcal{O}\left(\frac{N^2}{N}(|O|+|A|)\right)$ under homogeneous agents and worst-case 1-D link communication topology. We see that DG-MAPPO scales better in terms of cost complexity in such scenarios. Figure 4 shows cost comparison of DG-MAPPO against single-hop CTDE (ideal case), multi-hop CTDE.

### A.4.2 EXECUTION PHASE

CTDE methods are typically communication-free during execution: once training is complete, agents act based solely on local observations and fixed policies. However, this "silent" execution phase presupposes policies that were optimized under access to global state information. This creates a structural train–test mismatch that can severely degrade robustness under partial observability or dynamic topology changes that were not present during training.

DG-MAPPO, by design, uses the same communication mechanism during both training and execution. At inference time, agents continue to exchange feature vectors $h^i$ with neighbors via the D-GAT module. The per-timestep communication cost for a $K$-hop rollout is again given by Eq. 21. While this incurs a non-zero communication overhead during deployment, our ablation studies show that strong performance is already obtained with small hop budgets (e.g., $K = 1$ or $K = N/2$), so the execution-time communication cost remains bounded and tunable by design. In return, DG-MAPPO completely eliminates reliance on privileged global information, thereby avoiding the train-test mismatch inherent to CTDE.

### A.5 PHYSICAL COMMUNICATION COST AND DISTANCE

In real-world deployments, such as multi-robot teams or UAV swarms, communication cost is governed not only by bit rate but also by the physical distance over which signals are transmitted. The transmission energy $E_{\text{tx}}$ typically follows a power-law relationship with distance $d$,

$$E_{\text{tx}} \; \propto \; d^{\alpha}, \tag{25}$$

where $\alpha \in [2, 4]$ is the path-loss exponent determined by the propagation environment.

Let $b$ denote the encoded size (in bits) of a single transmitted embedding vector (e.g., a quantized version of $h^i$). For simplicity, we model the energy cost of transmitting $b$ bits over distance $d$ as

$$C(b, d) \; \approx \; b \, d^{\alpha}. \tag{26}$$

This abstraction is sufficient to compare the *relative* energy scaling of CTDE and DG-MAPPO.

**CTDE energy cost.** In CTDE, each agent must ultimately send its information to a central controller or parameter server. Let $d_{i,\text{center}}$ denote the (effective) distance between agent $i$ and the central learner, which may be realized via direct transmission or via multi-hop relaying. The total energy cost per update can be approximated as

$$E_{\text{CTDE}} \; = \; \sum_{i=1}^{N} C\big(b_i, d_{i,\text{center}}\big), \tag{27}$$

where $b_i$ is the number of bits sent by agent $i$ (e.g., encoding its observation, action, and possibly gradients). In large-scale environments, $d_{i,\text{center}}$ grows with the network diameter $D$, so the aggregate energy cost scales roughly as $\mathcal{O}\big(ND^{\alpha}\big)$.

**DG-MAPPO energy cost.** In DG-MAPPO, agents communicate only with physically adjacent neighbors within a limited communication radius $R$. Let $d_{i,j} \leq R$ be the distance between neighboring agents $i$ and $j$. The total energy cost per communication round can then be expressed as

$$E_{\text{DG-MAPPO}} \; = \; \sum_{i=1}^{N} \sum_{j \in N^i} C\big(b, d_{i,j}\big), \tag{28}$$

where $b$ is the bit-size of the transmitted embedding per link. For undirected links, each physical edge is counted twice in the double sum; one may divide by 2 if desired, but we retain the form in Eq. equation 28 for notational simplicity. Because $d_{i,j} \leq R$ by construction, the per-link energy cost is uniformly bounded, and for bounded node degree $|N^i|$ the total energy scales as $\mathcal{O}(N)$.

**Discussion.** Equations equation 27 and equation 28 highlight a key advantage of DG-MAPPO in physically realistic settings. CTDE requires long-range or multi-hop communication to a centralized learner, resulting in energy costs that increase with the network diameter and necessitating the maintenance of high-throughput long-range links. DG-MAPPO, in contrast, restricts communication to short-range neighbor exchanges whose cost is both *bounded* (via $R$) and *tunable* (via the hop budget $K$). Although DG-MAPPO incurs continuous communication during execution, these locally constrained transmissions result in more favorable energy scaling compared to the long-range communication required to support centralized learning in large and geographically extended multi-agent systems.

### A.6 LEARNING CURVES FOR CLIPPED SIGHT RANGE IN SMAC

Figures 5 and 6 demonstrate that DG-MAPPO remains effective even under sparse communication topologies, where the average node degree is approximately $N/2$. Moreover, our experiments reveal that respectable performance can be achieved even with 1-hop communication, and subsequent improvements in performance can be observed as the number of hops increases, albeit at the cost of slightly higher communication and computation overhead. Refer to Appendix X for an analysis of computation and communication overhead.

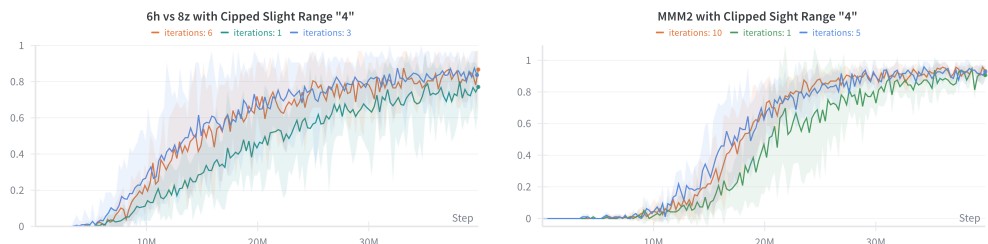

Figure 5: Evaluation win rate of DG-MAPPO under different hop counts with clipped sight range in the SMAC environment.

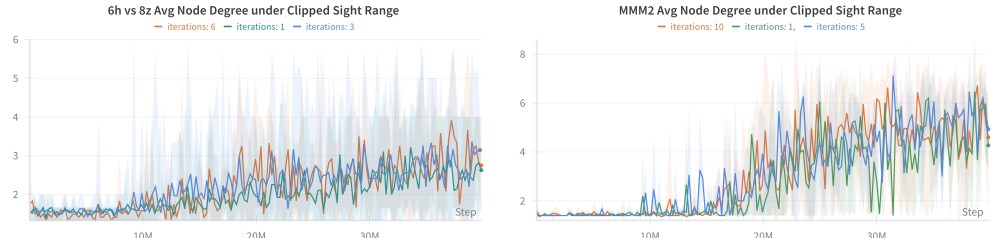

Figure 6: Average communication graph node degree at the final timestep of each SMAC episode with clipped sight range.

### A.7 ABLATION STUDY

The goal of our ablation study is to assess the importance of: 1) Attention mechanism for message aggregation, 2) number of communication hops on performance, and 3) consensus loss for performance stability. To test these components of DG-MAPPO, we strategically chose a "Hard" rated environment (*10m vs 11m*) with 10 collaborative agents, and two "Hard+" rated environments (*6h vs 8z* and *MMM2*) with 6 and 10 collaborative agents, respectively. We believe this provides us with sufficient diversity in terms of team size, heterogeneity and task complexity to effectively evaluate individual components of our algorithm.

#### A.7.1 MESSAGE AGGREGATION

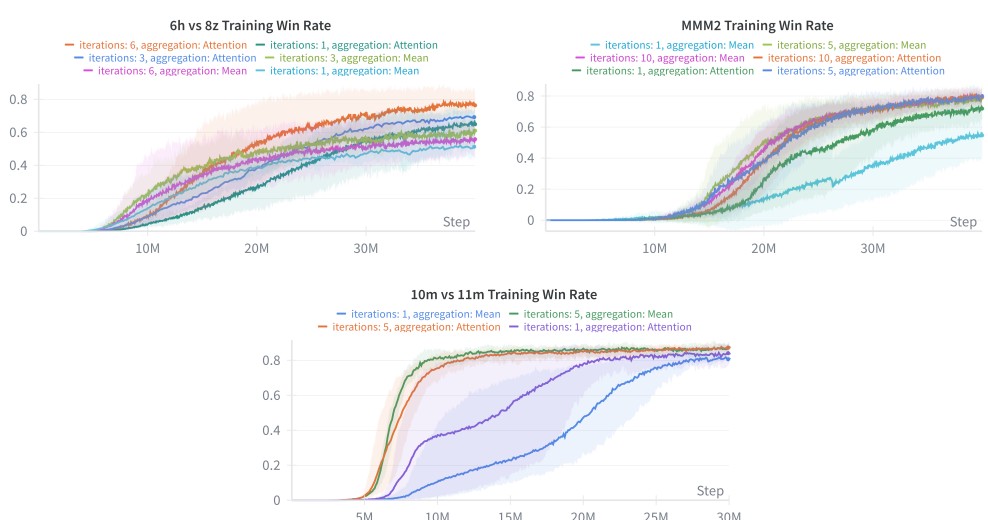

Figure 7: Training win rate comparison between "mean" message aggregation and attention-augmented aggregation across different hop counts (iterations) in SMAC environments.

Figure 7 compares mean-based and attention-augmented message aggregation within the D-GAT communication module. Although attention augmentation may appear inherently advantageous, its benefits turn out to be scenario-dependent. In the *6h vs 8z* setting—where the task is particularly challenging—attention-augmented aggregation consistently outperforms mean aggregation across all hop configurations. In contrast, for *MMM2* and *10m vs 11m*, mean aggregation achieves performance comparable to attention augmentation when the number of hops is set to $N/2$ or $N$, suggesting that one can reduce the computation cost of D-GAT without sacrificing performance in moderately complex scenarios. However, it is worth noting that in both environments, the 1-hop setting still favors attention-augmented aggregation, indicating its advantage when communication cost is the dominant concern.

### A.7.2 EFFECT OF NUMBER OF HOPS ON PERFORMANCE

We further evaluate DG-MAPPO under different choices of communication hops. Overall, increasing the number of hops generally improves performance, as broader information propagation enables better coordination. However, the trend is distinctly sublinear. As seen in Figures 3b and 7, DG-MAPPO already achieves strong performance with only 1-hop communication, and its results with $N/2$ hops are comparable to those of CTDE baselines that rely on global information. The marginal gains from increasing hops from $N/2$ to $N$ are present but relatively small. In practice, this suggests that one can begin with 1-hop communication and only increase the hop budget when the task difficulty or coordination demands justify the additional cost.

### A.7.3 EFFECT OF CONSENSUS LOSS ACROSS VARIOUS HOP CONFIGURATIONS

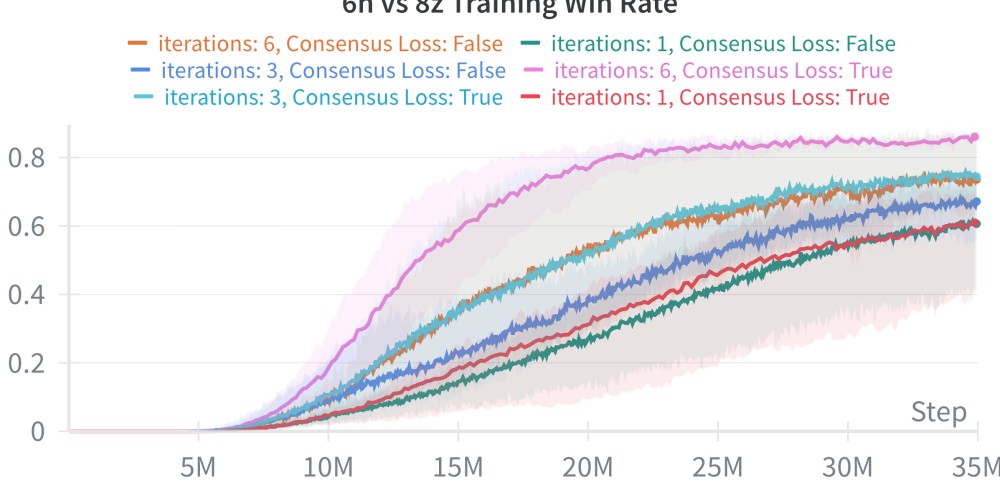

Figure 8: Training win-rate comparison in *6h vs 8z* scenario, illustrating the effect of the Consensus Loss under varying numbers of communication hops.

We further evaluate the impact of the proposed Consensus Loss under different communication hop budgets in the *6h vs 8z* scenario. As shown in Figure 8, the effect of this regularizer depends strongly on the available communication depth. When agents communicate with a relatively large hop budget (e.g., 6 hops), incorporating the Consensus Loss yields a substantial performance improvement. In this setting, messages propagate widely through the team, and enforcing agreement among neighboring representations accelerates the emergence of globally coherent policies. This is reflected in both faster learning and higher final win rates. In contrast, when the hop budget is restricted (e.g., 1 hop), the benefit of the Consensus Loss becomes more modest. Limited propagation constrains the ability of local consistency constraints to influence global coordination. Nevertheless, even in the 1-hop case, we observe a small but consistent improvement in learning speed, suggesting that encouraging local agreement still helps stabilize training under decentralized gradients. The intermediate 3-hop setting exhibits a clear improvement when the Consensus Loss is applied, although

the gain is smaller than what is observed with 6 hops. With three hops, agents can propagate information to a moderate portion of the team, allowing the regularizer to influence coordination more effectively than in the one-hop case. However, because information does not spread as widely as in the 6-hop configuration, the benefit of enforcing local agreement is correspondingly limited, leading to a moderate yet consistent performance increase. Overall, these results highlight that the Consensus Loss is most beneficial when communication is sufficiently expressive to carry its influence across the team. At the same time, the regularizer remains non-detrimental in low-communication regimes, supporting its use as a generally helpful stabilization mechanism in decentralized training.

## A.8  HYPER-PARAMETERS FOR DG-MAPPO

Table 4: Common hyper-parameters used for DG-MAPPO

| Parameter | Value | Parameter | Value | Parameter | Value |
|---|---|---|---|---|---|
| critic lr | 5e-4 | actor lr | 5e-4 | use GAE | True |
| gain | 0.01 | optim eps lr | 1e-5 | training threads | 32 |
| entropy coeff | 0.001 | max grad norm | 10 | optimizer | Adam |
| hidden layer dim | 128 | use huber loss | True | gae lambda | 0.95 |
| D-GAT lr | 5e-4 | num heads | 1 | $\alpha_{consensus}$ | 20 |

Table 5: Different hyper-parameters used for DG-MAPPO

| Maps | ppo epochs | ppo clip | batch size | rollout threads | episode length | gamma | steps |
|---|---|---|---|---|---|---|---|
| 3m | 10 | 0.05 | 2048 | 32 | 100 | 0.98 | 2e6 |
| 8m | 10 | 0.05 | 2048 | 32 | 300 | 0.98 | 1e7 |
| MMM | 10 | 0.05 | 2048 | 32 | 300 | 0.98 | 1e7 |
| 5m vs 6m | 5 | 0.05 | 3200 | 32 | 300 | 0.98 | 3e7 |
| 8m vs 9m | 10 | 0.05 | 2048 | 16 | 500 | 0.98 | 2e7 |
| 10m vs 11m | 5 | 0.05 | 3200 | 32 | 500 | 0.98 | 2.5e7 |
| 25m | 5 | 0.05 | 4800 | 16 | 300 | 0.95 | 1e7 |
| MMM2 | 5 | 0.05 | 3200 | 32 | 300 | 0.95 | 3e7 |
| 6h vs 8z | 10 | 0.05 | 2048 | 32 | 300 | 0.98 | 2e7 |
| 3s5z vs 3s6z | 5 | 0.2 | 3200 | 32 | 300 | 0.98 | 3e7 |

### A.8.1  DG-MAPPO EVALUATION PLOTS

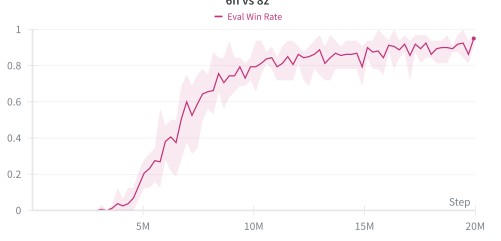
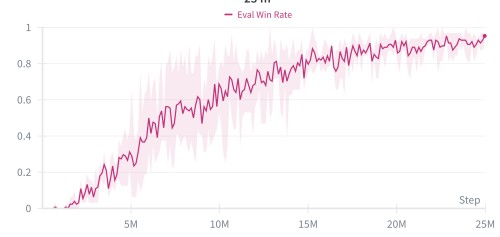

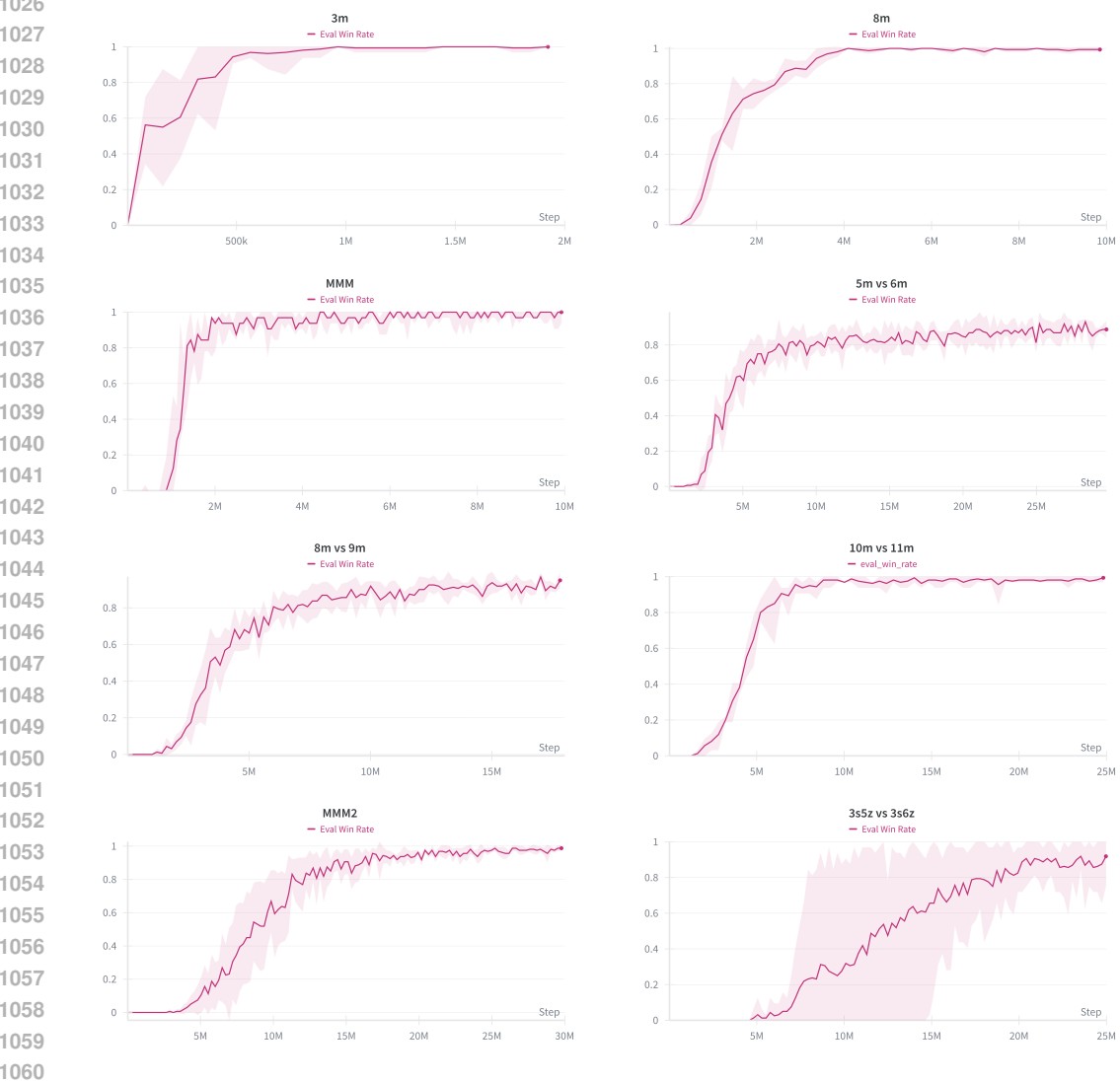

## A.9    AVERAGE NODE DEGREE (AT END OF EPISODE)

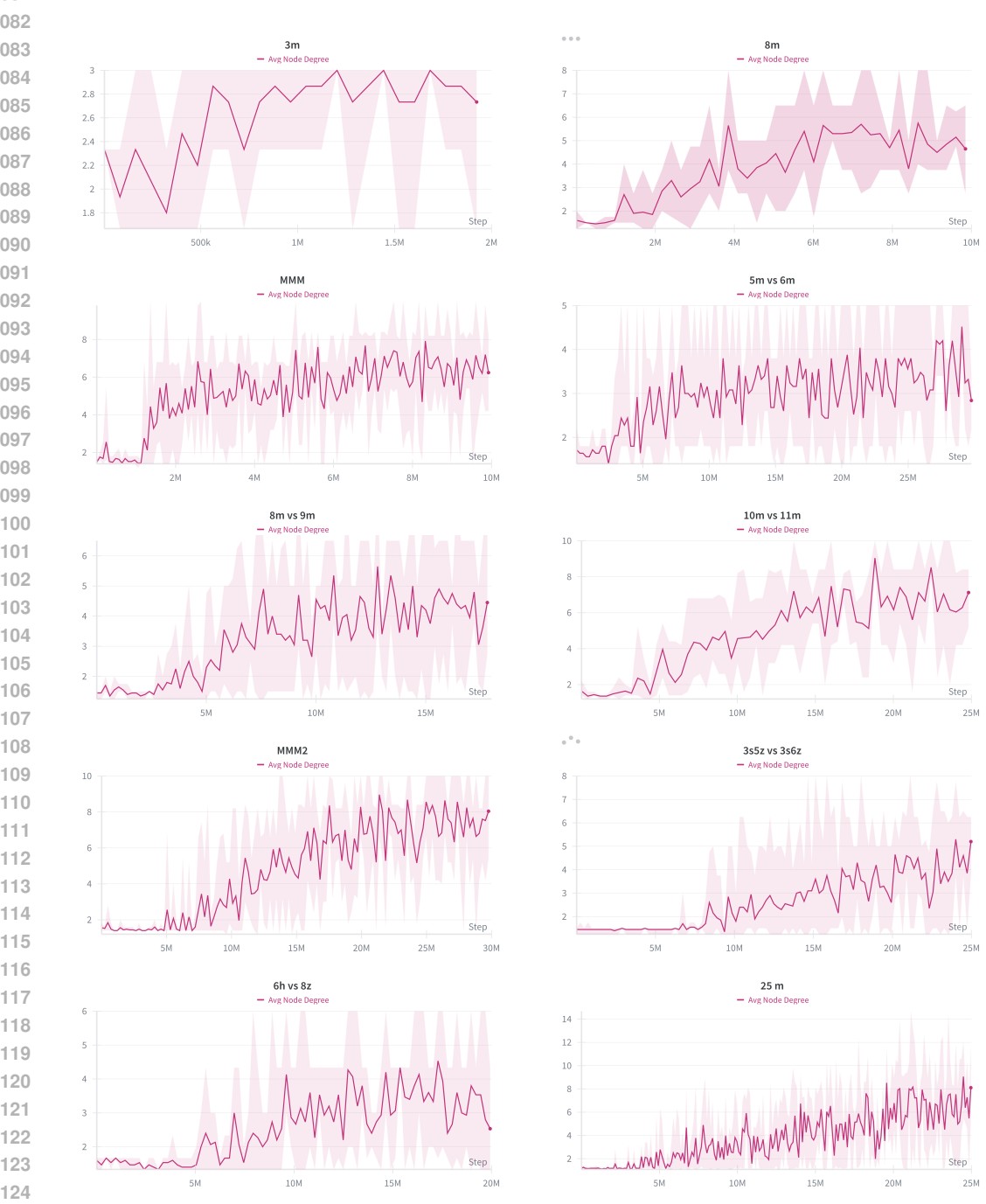