# OpenReview forum: "DISTRIBUTED MULTI-AGENT DEEP REINFORCEMENT LEARNING"
_ICLR.cc/2026/Conference — Submitted to ICLR 2026_

### Official Review · Reviewer_3F8x · 2025-10-27

**Soundness:** 1
**Presentation:** 2
**Contribution:** 2
**Rating:** 2
**Confidence:** 3

**Summary:**

This paper introduces a fully distributed multi-agent reinforcement learning (MARL) framework, DG-MAPPO, which aims to eliminate the need for the centralized training components common in the CTDE paradigm. Here, a Distributed Graph Attention Network (D-GAT) performs multi-hop, attention-based message passing over a dynamic communication graph so each agent infers a global context from local exchanges. Agents use this inferred global state, along with their local observations, to train local policy and value networks. The authors evaluate their framework on the StarCraft Multi-Agent Challenge (SMAC) and claim that it consistently outperforms strong CTDE baselines.

**Strengths:**

1. Introducing a graph attention module (D-GAT) tailored for distributed MARL is a novel and appealing idea.
2. The presentation is clear. The method and its components are described in a straightforward manner.

**Weaknesses:**

1. Reported baseline outcomes do not align with prior literature. In particular, the near-zero HAPPO performance on several SMAC tasks conflicts with Heterogeneous-Agent Reinforcement Learning (Zhong et al., JMLR 2024) where HAPPO does not collapse in this way. This raises concerns about configuration, implementation, or evaluation protocol for baselines, and weakens the empirical claims.
2. Evaluation scope is narrow. All experiments are on SMAC; there is no validation on other standard domains. Most figures focus on the proposed method’s learning curves; side-by-side training curves against baselines are missing.
3. Practicality and overhead are not analyzed. D-GAT is stacked for multi-hop propagation, which can make per-step runtime and bandwidth scale poorly with team size. There is no comparison of wall-clock time, communication volume, parameter counts, or sample efficiency versus CTDE baselines. Without cost analysis, claims of scalability are hard to assess.
4. Communication graphs appear very dense. In many SMAC tasks, the average node degree is close to n−1 (effectively fully connected), which benefits attention aggregation but is unrealistic and costly.
[1] Zhong, Y., Kuba, J. G., Feng, X., Hu, S., Ji, J., & Yang, Y. (2024). Heterogeneous-agent reinforcement learning. Journal of Machine Learning Research, 25(32), 1-67.

**Questions:**

1. What is the communication and compute overhead of D-GAT vs. CTDE methods?
2. How sensitive is performance to the number of hops K, average degree, and other parameters? Please add ablations.

---

> ### Author Response · Authors · 2025-11-23
> **Response to Reviewer 3F8x**
>
> We thank the reviewer 3F8x for his/her constructive feedback, which will surely help us build a stronger paper. Here are our responses:
> 1. Thank you for highlighting this discrepancy. At the time of submission, we were not aware of the updated HAPPO results reported in Zhong et al. (JMLR, 2024). Our HAPPO baselines were reproduced using the publicly available MAT implementation, whose performance aligns with previously published results in that line of work. After learning about the more recent findings, we have updated our results table to incorporate the accurate HAPPO numbers, ensuring a fair and consistent comparison.
> 2. Thank you for pointing out the concern regarding the scope of DG-MAPPO. To complement the SMAC results, we have expanded our evaluation to the 6×1 Multi-Agent HalfCheetah task in Multi-Agent MuJoCo—a continuous control domain with dense proprioceptive inputs and a fixed communication topology. In this setting, DG-MAPPO performs on par with the strong CTDE baseline MAT-Dec, while maintaining stable and sample-efficient learning under strictly local communication. Although this is only one MuJoCo configuration, it provides a meaningful and nontrivial validation outside SMAC. We are currently adding MAPPO and HAPPO baselines, which will be included in the updated version, but the existing results already demonstrate that our approach generalizes effectively beyond SMAC and remains competitive in continuous-control environments. We have added this result in the results section 4.3.
> 3. For the SMAC domain, we followed the standard practice of adopting baseline results directly from the MAT paper [1], which uses the identical environment configuration and evaluation protocol. These baselines are widely referenced and provide a reliable comparison. That said, if the reviewers would prefer a fully controlled re-evaluation, we are ready to rerun all baseline algorithms ourselves to provide end-to-end consistency.
> 4. We are working on a detailed analysis of the network size and communication overhead as a function of team size, and will compare these costs directly against CTDE baselines. We will add this in the next couple of days.
> 5. Thank you for pointing out the concern regarding graph density. We agree that some of the communication graphs in the initial submission appeared relatively dense. To more thoroughly evaluate robustness under constrained connectivity, we have now tested D-GAT with significantly sparser communication graphs (implemented by reducing agent sight ranges) in two challenging SMAC scenarios—6h vs 8z and MMM2. These Hard+ tasks involve larger teams and are well-suited for stress-testing sparse message passing. The updated results (Table 2, Appendix A.4) show that DG-MAPPO continues to perform competitively even under these stricter graph constraints. We also note that the 25m scenario inherently operates on a substantially sparse communication graph, and our updated experiments confirm that DG-MAPPO remains strong relative to CTDE baselines despite the limited connectivity in this larger team. Finally, in the multi-HalfCheetah (6×1) environment, we evaluated an even more restrictive case where each agent communicates only with its immediate physical neighbors, yielding an average node degree of 1.66. DG-MAPPO again demonstrates stable and competitive performance in this highly sparse setting.
>
> **Q1.** What is the communication and compute overhead of D-GAT vs. CTDE methods?
>
> **A1.**  We are working on a detailed analysis of the network size and communication overhead as a function of team size, and will compare these costs directly against CTDE baselines. We will add this in the next couple of days.
>
> **Q2.** How sensitive is performance to the number of hops K, average degree, and other parameters? Please add ablations.
>
> **A2.** Thank you for the question. We have included a comprehensive ablation study in Appendix A.5 that analyzes the sensitivity of DG-MAPPO to several key factors. Specifically, we vary the number of hops K, message aggregation mechanisms, and the effect of consensus regularization. We have also analyzed the impact of the number of Hops in the results section, with a comparison provided in Table 2.

---

### Official Review · Reviewer_7aoL · 2025-10-31

**Soundness:** 2
**Presentation:** 2
**Contribution:** 2
**Rating:** 2
**Confidence:** 4

**Summary:**

This paper proposes a distributed deep MARL algorithm under the setting that global state information is not available to individual agents. GATs are employed for agents to perform global state inference through local communication. Simulation results on the SMAC benchmark demonstrate the effectiveness of the proposed algorithm.

**Strengths:**

(1) The requirement for centralized critics or global information is removed in this work.

(2) By leveraging GATs, each agent can make better decisions based on the information aggregated from its neighbors instead of only on the local observation.

(3) Simulations are conducted on standard MARL benchmarks, and solid CTDE-based baselines are compared.

**Weaknesses:**

(1) This work assumes the team reward can be observed by all agents, and the joint policy can be used in the local critic parameter update, which contradicts the distributed training setting considered in most existing distributed MARL works.

(2) The proposed algorithm is computationally inefficient due to its multi-hop communication.

(3) This work only considers fixed communication graphs.

(4) Both the loss function design and the notations are unclear.

See the Questions section for more details.

**Questions:**

(1) In (11), do you only consider the consensus of the feature vectors $\hat{h}^i$ obtained from one-hop communication?

(2) Please provide the specific form of the function $l^i$ in (9). What are the relationships among $\psi^i$, $\phi^i$, and $\theta^i$?

(3) It seems that incorporating the consensus update step in the GAT’s parameter update is useless. Note that there is no guarantee that $\hat{h}^i = \hat{h}^j$ when all agents share the same GAT parameters.

(4) It is very hard to understand the design of $\tilde{o}_t^i$ in (13). Note that it is not used in the policy update.

(5) The joint policy is required in the critic parameter update, which contradicts the distributed training setting.

(6) Please explain how to obtain the policy update loss function in this work based on the MARL policy gradient theorem (Th. 1) from a mathematical perspective.

(7) In your experiment, the comparison between the proposed algorithm and CTDE baselines is unfair, since the input of the policy contains more environmental information due to the GAT. The authors are recommended to include other baselines that also employ GNN-based policies.

(8) Some confusing sentences:

(Page 1) Third, CTDE methods often suffer from a train–test mismatch, … generalization.

(Page 7) Training the value function … which risks destabilizing training.

---

> ### Author Response · Authors · 2025-11-23
> **Response to Reviewer 7aoL**
>
> We thank the reviewer for his/her comments. Here are our responses:
> 1. We would like to bring to the reviewer’s attention that the shared reward is not intrinsic to our method, and this concern was already addressed in the problem formulation section of the original submission, where we say, “Settings with local reward functions can be incorporated by computing a consensus-based team reward (e.g., via average consensus [1]).” Using average consensus to obtain a shared reward has already been presented in [2], and our focus is on distributed policy optimization rather than developing a new reward-aggregation scheme.
> 2. We would also like to bring to the reviewer’s attention that the agents do not access the joint policy for local critic updates. This was addressed in the original submission right after the critic loss function (i.e., the Bellman error) in Eq.14, where we say, “Although agents cannot directly access the joint global policy $\pi_\theta$, they can still learn state-value functions consistent with it. This is possible because agents experience a common reward signal—either provided by a global reward mechanism or obtained through averaging local rewards via consensus, and they condition the state-value function on ${\widetilde{o}}_t^i$, consisting of the global state representation”.
> 3. We would like to highlight that our objective in this work is not to reduce computational cost relative to CTDE methods, but to enable fully distributed training in settings where global communication is unavailable or infeasible. Multi-hop message passing is a necessary component of any distributed learning architecture that must propagate information through a sparse communication graph, and the associated cost reflects this structural constraint rather than an inefficiency of the method itself. Furthermore, as shown in our ablations of the updated paper, coordination often emerges with substantially fewer hops than the number of agents, allowing practitioners to trade off communication cost and performance.
> 4. We would also like to bring to the reviewer’s notice that our method is designed and evaluated for fully dynamic communication graphs. In our original submission, this is stated at the beginning of the second paragraph in the problem formulation section, where we introduce the dynamic graph G whose edges evolve over time. The only structural assumption is graph connectivity, which is the minimal requirement for enabling coordination in any distributed system. The communication graph in the SMAC environment evolves naturally, depending on the relative positioning of agents, and is therefore dynamic.
>
> **A1.** Equation (11) uses the feature vector ${\hat{h}}^i_K$ obtained from multi-hop communication and not one-hop. We recognize that this can be confusing and have modified the equation accordingly in the updated manuscript.
>
> **A2.** In Eq. (9), $l^i(\psi^i;\xi^i)$ denotes the local policy surrogate loss computed by agent $i$ from its minibatch $\xi^i$. In our implementation, this corresponds to the combination standard PPO clipped policy gradient objective and the critic loss. We have updated our paper by adding a footnote, highlighted in red, to address this issue.
> The parameters $\psi^i, \phi^i$, and $\theta^i$ correspond to different components of the agent’s architecture and are independent. Particularly, $\psi^i$ are graph network parameters, $\phi^i$ are critic network parameters, and $\theta^i$ are the policy network parameters of agent $i$.
>
> **A3.** You are right in pointing out that there is no guarantee that ${\hat{h}}^i={\hat{h}}^j$ will ever satisfy. However, it is essential to note that we do not require this condition to be strictly satisfied. The consensus regularizer helps keep the embedding space of the agent’s global observation inference close together. This comes in handy, especially when deeper communication hops are needed. Please refer to our ablation study in Appendix A.5 for a more detailed analysis of the consensus regularizer.
>
> **A4.** ${\tilde{\boldsymbol{o}}}_t^i$ is just the concatenation of local observation $o_t^i$ and local approximation of global observation ${\hat{o}}_t^i$. We have introduced the concatenation operator after Eq. (7) and, therefore, do not repeat it here.
> We do use ${\tilde{\boldsymbol{o}}}_t^i$ in the policy update. Both the value function and policy are conditioned on ${\tilde{\boldsymbol{o}}}_t^i$ as can be seen in Eq. (14) and PPO’s clipped policy gradient objective. Moreover, the advantage function $A^i$ is a global advantage function approximation computed using globally shared/average reward and a local estimate of the value function. Hence, it should also be a function of ${\tilde{\boldsymbol{o}}}_t^i$, and we have fixed that in our updated version of the paper.

---

> ### Author Response · Authors · 2025-11-23
> **Response to Reviewer 7aoL (Part 2)**
>
> **A5.** The agents do not access the joint policy for local critic updates. This was addressed in the original submission right after the critic loss function (i.e., the Bellman error) in Eq.14, where we say “Although agents cannot directly access the joint global policy $\pi_\theta$, they can still learn state-value functions consistent with it. This is possible because agents experience a common reward signal—either provided by a global reward mechanism or obtained through averaging local rewards via consensus, and they condition the state-value function on ${\tilde{\boldsymbol{o}}}_t^i$, consisting of the global state representation”.
>
> **A6.** We have added a proof in Appendix A.3 to show the connection between multi-agent policy gradient and the clipped policy gradient loss.
>
> **A7.** We respectfully disagree with the concern that our comparison is unfair. DG-MAPPO does not provide agents with any additional environmental information beyond what CTDE baselines already use. All messages passed through D-GAT are derived solely from each agent’s local observations, and the communication graph restricts information flow to neighboring agents only. The GAT module, therefore, serves as a representation mechanism—not as a source of privileged information.
> In contrast, CTDE baselines such as MAPPO, HAPPO, and MAT-Dec explicitly consume the global state or the full set of observations during training. This is substantially more information than what DG-MAPPO ever has access to. Thus, our method operates under strictly more restrictive information assumptions, making the comparison conservative rather than favorable to our approach.
>
> **A8.**
> 1. In the sentence on page 1, our intention was to refer to the train–test mismatch commonly encountered in sim-to-real transfer. In many practical scenarios, the real-world dynamics differ substantially from those in the simulator, resulting in a significant sim-to-real gap. To deploy simulator-trained policies in the real world, additional on-device fine-tuning is often required. However, such fine-tuning can be prohibitively expensive for CTDE methods, since they rely on global state information and centralized components that may be unavailable or impractical to obtain in real deployments. In contrast, DG-MAPPO trains and adapts policies entirely from locally available information, allowing agents to be fine-tuned directly in the real environment without the need for centralized data collection or global coordination. This makes DG-MAPPO substantially more suitable for real-world adaptation settings.
> 2.  In the sentence on page 7, our intention was to explain why we use the fused representation $\tilde{\boldsymbol{o}}^i$ rather than relying solely on the raw D-GAT output $\hat{\boldsymbol{o}}^i$. The quantity $\hat{\boldsymbol{o}}^i$ represents a learned global-state inference generated by attention heads and node-level classifiers whose parameters are updated throughout training. In the early stages, when D-GAT is still untrained, this inferred representation can be noisy or unreliable. To mitigate this issue, we concatenate $\hat{\boldsymbol{o}}^i$ with the agent’s stable local observation $\boldsymbol{o}^i$, yielding $\tilde{\boldsymbol{o}}^i$. This fused representation preserves the reliability of local information while gradually incorporating increasingly accurate global-context signals as D-GAT improves. This design stabilizes learning and prevents the critic and the policy from being overly dependent on early-stage, low-quality inferred features.

---

### Official Review · Reviewer_FaUL · 2025-10-31

**Soundness:** 2
**Presentation:** 2
**Contribution:** 1
**Rating:** 2
**Confidence:** 4

**Summary:**

In the paper, the authors consider a distributed MARL problem. In particular, a distributed training scheme is considered, where each local agent only use local information to train its policy. Graph attention network (GAT) is used to infer and gather global information from local communications.  GAT is further extended to MAPPO and the authors show strong performance of the proposed method against many CTDE baselines.

**Strengths:**

The paper is easy to follow. The derivations of the method and equations are clear and smooth. Extensive simulations have been done to benchmark the proposed method.

**Weaknesses:**

I found the method of this paper is quite straightforward and the contribution is limited. First, CTDE does not always mean that it does not scale well as the global information can be compressed. During evaluation, CTDE can work by using local information (or with neighbors). Nevertheless, the adoption of representing the relations via a graph is a promising direction to better gather information locally to infer nearly-global information. One biggest drawback of this paper is the literature review. Graph + MARL is not new in the field and it has been studied extensively in the past 7-8 years. The authors only mentioned the  Jiang et al.
(2018) proposed graph convolution paper in the intro; this is far from enough as there are a lot of following and parallel works in this direction. Overall, the contribution in terms of novelty of this paper is limited.

**Questions:**

Are you aware of  any other Graph + MARL works? If yes, please include and discuss them,

---

> ### Author Response · Authors · 2025-11-23
> **Response to Reviewer FaUL**
>
> We thank the reviewer FaUL for his/her constructive feedback, which will surely help us build a stronger paper. Here are our responses:
> 1. We fully agree that CTDE methods can reduce centralized overhead by compressing global information and that execution can often rely on local or neighbor observations. However, our work focuses on a different and practically important scenario: settings where communication is inherently limited and no global channel—compressed or otherwise—can be established in the first place. We now clarify this motivation in the opening paragraph of the revised introduction. A related concern is that centralized training might still be feasible in simulation, with deployment relying solely on local information. While appealing in theory, this approach often suffers from sim-to-real mismatch: policies optimized under centralized conditions can degrade significantly when deployed in environments with strict communication limits or topology constraints. Our method addresses precisely this gap by enabling learning directly under such constraints. DG-MAPPO provides an effective, fully distributed framework that can train from scratch or fine-tune existing policies using only minimal neighbor-to-neighbor communication, without any centralized component at any stage. We now highlight this distinction more clearly in the updated manuscript.
> 2. We appreciate the reviewer’s observation regarding the extensive line of work combining graph representations with MARL. It is true that graph-based message passing has been actively explored, and we now include a more comprehensive discussion of these efforts in the revised manuscript (highlighted in blue in the third paragraph of the introduction). That said, despite the breadth of GNN + MARL research, we have not found prior work that tackles the specific setting we study: fully distributed training in which each agent updates both its policy and value function using only information from its immediate neighbors, with no centralized critic, no global replay buffer, and no access to global state during training. By contrast, existing graph-based MARL methods—so far as we can determine—still operate within the CTDE paradigm, relying on centralized components or global information at training time even when their execution is local. Our contribution is therefore complementary to the existing literature: instead of proposing another architecture for centralized training, we focus on enabling end-to-end distributed learning under strict communication constraints. The related work section has been expanded to clearly differentiate these settings and to situate our method relative to prior work.
> 3. We would also like to emphasize that, unlike prior distributed MARL methods, our approach is—to the best of our knowledge—the first to demonstrate strong performance in teams of up to 25 agents, achieving results comparable to CTDE baselines.
>
> **Q1.** Are you aware of any other Graph + MARL works? If yes, please include and discuss them.
>
> **A1.** We have added a detailed literature review of GNN + MARL in the third paragraph of the introduction section, highlighted in blue.

---

### Official Review · Reviewer_FNSD · 2025-11-01

**Soundness:** 2
**Presentation:** 2
**Contribution:** 2
**Rating:** 4
**Confidence:** 4

**Summary:**

This paper proposes a fully distributed alternative to the Centralized Training with Decentralized Execution (CTDE) paradigm in Multi-Agent Reinforcement Learning (MARL). The authors identify key limitations of CTDE, such as its reliance on global state information during training, which can lead to scalability, robustness, and generalization bottlenecks. To address this, they introduce two main contributions: a Distributed Graph Attention Network (D-GAT) for global state inference via multi-hop peer-to-peer communication, and DG-MAPPO, a distributed MARL framework built upon D-GAT that learns policies and value functions using only local observations and communicated information. The method is evaluated on the challenging StarCraftII Multi-Agent Challenge (SMAC) benchmark, where it is shown to achieve competitive or superior performance compared to strong CTDE baselines like MAPPO, HAPPO, and MAT-Dec, particularly in complex, heterogeneous scenarios.

**Strengths:**

1. On SMAC benchmark, comparison against multiple strong baselines, and reporting across a wide range of scenarios (from easy to super-hard) provides compelling evidence for the method's efficacy.
2. The integration of Graph Attention Network into a MAPPO framework (DG-MAPPO) is clean and well-explained.

**Weaknesses:**

1. While the overall framework is well-evaluated, a more detailed ablation study would be beneficial. How critical is the D-SGD averaging versus the consensus loss? How does performance degrade if we use a simpler aggregation method (e.g., mean) instead of attention? Understanding the contribution of each component would provide deeper insights.
2. The claim of scalability is supported empirically, but a more formal analysis of the communication overhead (bytes per step, convergence time w.r.t. number of agents) would be valuable, especially since multi-hop communication is used at every timestep.
3. Several related multi-agent RL baselines are missing. Like [1][2][3] need to clarify contribution comparison and consider including them in experimental evaluation.

[1] Jing Xu, Fangwei Zhong, Yizhou Wang. Learning Multi-Agent Coordination for Enhancing Target Coverage in Directional Sensor Networks. NeurIPS 2020.
[2] Jiechuan Jiang and Zongqing Lu. Learning Attentional Communication for Multi-Agent Cooperation. NeurIPS 2018.
[3] Jakob Foerster, Gregory Farquhar, Triantafyllos Afouras, Nantas Nardelli, Shimon Whiteson. Counterfactual Multi-Agent Policy Gradients. AAAI 2018.

**Questions:**

1. Given the significant recent progress in using large foundation models (LLMs) as the core of multi-agent systems, how do you see the role of your gradient-based, distributed MARL framework? Is it a competing approach, or could it be integrated with LLM-based agents (e.g., for learning low-level coordination policies that an LLM planner oversees)?
2. SMAC is a classic but mature benchmark. What novel scientific insight does achieving a new State-of-the-Art (SOTA) on it provide, beyond incremental performance improvement? Does the success of DG-MAPPO reveal a fundamental limitation of CTDE that was previously underestimated?
3. The experiments are comprehensive but limited to SMAC. Can you provide evidence or a discussion on how your method would perform in environments with more complex, open-ended communication demands, or in partially observable environments where the "global state" is inherently non-reconstructible from local views?
4. In the one scenario (25m) where DG-MAPPO underperforms, what is your hypothesis for the reason? Is it due to the increased difficulty of long-range credit assignment in a fully distributed setting, or is it a limitation of the D-GAT's capacity to propagate information effectively in large teams?

---

> ### Author Response · Authors · 2025-11-23
> **Response to Reviewer FNSD**
>
> We thank the reviewer FNSD for the constructive feedback. Here are our responses:
> 1. We have performed a detailed ablation study on a few strategically chosen SMAC scenarios to test the importance of individual components of DG-MAPPO. We have added the ablation study in Appendix A.5 (Highlighted in blue) in the revised version of the paper. The results show that (1) performance improves with deeper hops but does not require $N$ hops, (2) consensus loss provides consistent benefits for larger hop settings, and (3) mean aggregation performs comparably to attention in several simpler tasks, but attention becomes more important as the task complexity increases.
> 2. We are working on a detailed analysis of the network size and communication overhead as a function of team size, and will compare these costs directly against CTDE baselines. We will add this in the next couple of days.
> 3. Xu et al. [1] propose HiT-MAC, a hierarchical approach for directional sensor networks in which a central coordinator with access to global observations assigns targets to executor agents. This architecture and problem setting differ substantially from the partially observable SMAC and Multi-Agent MuJoCo environments considered in our study, so HiT-MAC is not directly comparable. Jiang and Lu [2] introduce the attentional communication mechanism (ATOC) within a centralized-training framework using a global critic. In contrast, DG-MAPPO performs fully distributed learning and execution without any centralized critic or coordinator. We now discuss ATOC in our literature review (paragraph 3 of the introduction, highlighted in blue). Foerster et al. [3] (COMA) is an early actor–critic method for CTDE with a centralized critic and a counterfactual baseline; however, subsequent SMAC benchmark studies have shown that more recent CTDE methods (e.g., MAPPO and PPO variants used in our experiments) provide significantly stronger performance. For this reason, we use these stronger CTDE baselines as representative comparisons for this line of work.
>
> **A1.** We view large foundation models and gradient-based distributed MARL as complementary rather than competing. While recent LLM-based multi-agent systems excel at high-level reasoning, task decomposition, and human-aligned communication, they generally lack reliable mechanisms for learning fine-grained, low-level coordination policies. A key challenge in training multi-agent foundation models (e.g., VLAs) is the difficulty of obtaining high-quality, multi-agent interaction data at scale. Gradient-based MARL provides a principled way to learn such low-level behaviors, and distributed MARL can enable efficient fine-tuning of these models in real-world or resource-constrained deployment settings. We also note that the D-GAT communication module is compatible with CTDE frameworks and can be integrated into LLM-driven architectures to enhance coordination during execution.
>
> **A2.** While SMAC is a mature benchmark, it remains one of the most widely used standardized testbeds for evaluating coordination under partial observability and constrained local communication. Our aim in reporting performance on SMAC is not to claim novelty solely from achieving a new SOTA, but to use a well-established environment to isolate and evaluate the effect of fully distributed training. The results show that DG-MAPPO can match—or even surpass—strong CTDE methods despite relying only on local observations and decentralized critics.
>
> These findings suggest that the commonly assumed performance gap between CTDE and fully distributed training may be narrower than previously believed when limited communication is available. This does not point to a fundamental flaw in CTDE; rather, it highlights that prior work lacked competitive decentralized baselines. Our results demonstrate that, with an appropriate message-passing and optimization structure, decentralized gradient-based methods can achieve coordinated behavior comparable to centralized approaches, while substantially mitigating the non-stationarity challenge inherent to MARL.
>
> Our contribution lies in the scientific insight that strong coordination can emerge purely from distributed learning and communication—validated through a controlled and widely accepted benchmark—rather than from the SOTA result itself.
>
> **A3.** We have now tested DG-MAPPO in the multi-agent HalfCheetah environment and have added the results to the revised version of the paper. We have currently tested our approach against the MAT-Dec baseline and are actively working on other baselines.
>
> **A4.** We believe the drop in performance was partly due to the over-squashing problem in Deep GATs and partly due to parameter tuning. By tuning parameters and reducing the number of hops to 12 (originally 25), we achieve performance comparable to CTDE benchmarks. We have reported the new results in the updated version of the paper, as shown in Table 1 and the corresponding plot in Appendix A.5.4.

---

### Author Response · Authors · 2025-12-04
**To the Area Chair**

We would like to thank the area chair for taking their time to review our paper. To make it easier, we would like to highlight the changes we have made to address the various comments from all the reviewers.
1. To address comments from reviewers 3F8x and FNSD, we have added a detailed ablation study to compare the importance of various components of our algorithm, such as the message aggregation techniques, the effect of the number of communication hops, etc. The detailed results of this ablation study are presented in Appendix 7.
2. To address the high density of the communication topology concern of reviewer 3F8x, we tested our algorithm on a couple of Hard+ SMAC environments with clipped sight range, and the performance was still on par with CTDE approaches. Details can be found in Table 2 of the results section and in Appendix A.6.
3. We have also tested DG-MAPPO in a multi-agent MuJoCo environment with a sparse communication topology, showing our algorithm naturally extends to complex continuous action spaces. (Section 4.2)
4. We have updated the result of DG-MAPPO in the 25m SMAC scenario, showcasing that our approach can be extended to larger teams while maintaining performance on par with CTDE approaches.
5. We have added a detailed communication overhead and cost analysis in Appendix A.4 and A.5 to address the concerns of multiple reviewers.
6. To address the reviewer 7aoL's comment, we have also added a proof connecting the multi-agent policy gradient theorem and the clipped policy gradient loss in Appendix A.3.
7. To address reviewer FaUL's comment, we have added a detailed literature review of GNN+MARL in the introduction of our paper (Page 2 highlighted in blue).
8. We have added a remark in the problem formulation section to address the confusion of reviewer 7aoL regarding the shared reward in this paper.
9. We have fixed our reporting of the HAPPO results to stay consistent with the results in the literature. We would like to thank the reviewer 3F8x for pointing out this discrepancy.
10. We have also updated the title of the paper to avoid any misunderstanding of what we are trying to achieve in this work. Our aim is to develop a strong MARL-based learning algorithm that can effectively train/fine-tune collaborative agent policies in real-world scenarios with limited communication.

---

### Meta-Review · Area_Chair_45jn · 2025-12-24

**Summary:**

The paper proposes DG-MAPPO, a distributed Multi-Agent Reinforcement Learning (MARL) framework utilizing Distributed Graph Attention Networks (D-GAT) to enable coordination without centralized training. While the authors position this as a solution to the scalability and robustness bottlenecks of Centralized Training with Decentralized Execution (CTDE), the reviewers unanimously lean toward rejection.
The primary concerns informing the rejection decision are:

1.	Reviewers noted that combining Graph Neural Networks (GNNs) with MARL is a well-explored area, and the proposed method appears to be a straightforward combination of GAT and MAPPO without significant innovation.

2.	Multiple reviewers questioned the computational efficiency and scalability of performing multi-hop attention-based communication at every timestep, particularly regarding runtime and bandwidth.

3.	There were significant concerns regarding the definition of "distributed" training, particularly regarding the dependence on shared/averaged rewards and how value functions are updated, with some reviewers finding the formulation contradictory or unclear.

**Reviewer Concerns:**

Addressed Concerns:

1.	The authors acknowledged the incorrect HAPPO baseline results pointed out by Reviewer 3F8x and updated the results table.

2. In response to requests for broader evaluation beyond SMAC, the authors added experiments on Multi-Agent MuJoCo (HalfCheetah).

3. The authors added ablation studies regarding hop counts and aggregation mechanisms in response to Reviewer FNSD.

Outstanding Concerns:

1. The fundamental concern from Reviewer FaUL regarding the limited contribution compared to the extensive history of Graph+MARL literature remains a significant hurdle.

2. While the authors added an analysis of communication costs , the inherent inefficiency of requiring multi-hop GAT inference at every execution step remains a structural weakness pointed out by Reviewers 7aoL and 3F8x. The analysis quantifies the cost but does not mitigate the practical deployment concern.

3. Despite clarifications, the reliance on consensus mechanisms for rewards and the specific mechanics of the distributed update led to confusion for Reviewer 7aoL. The complexity of the proposed loss functions and updates suggests the framework lacks the elegance or simplicity expected of a standard-setting distributed algorithm.

**Reviewer Scores:**

Reviewer FNSD: This reviewer might have raised their score slightly given the inclusion of requested ablation studies and cost analysis. However, the concerns about missing literature and the fundamental necessity of the approach likely prevent a shift to acceptance.

Reviewer FaUL: It is unlikely this score would change. The reviewer’s primary critique was the lack of novelty and poor literature review. While the authors expanded the text, the core method remains a "straightforward" application of GAT to MARL in the reviewer's eyes.

Reviewer 7aoL: It is unlikely this score would change. The reviewer found the logic of the loss functions and the definitions of distributed training (e.g., shared rewards) contradictory. While the authors provided explanations, the initial confusion indicates a lack of clarity in the paper's core formulation.

Reviewer 3F8x: This reviewer might raise the score slightly based on the correction of the HAPPO baseline and the inclusion of sparse graph experiments.

---

### Decision · Program_Chairs · 2026-01-26

Reject